# Model-Based Diffusion for Trajectory Optimization

**Chaoyi Pan**[*], **Zeji Yi**[*], **Guanya Shi**[†], **Guannan Qu**[†]
Carnegie Mellon University
{chaoyip,zejiy,guanyas,gqu}@andrew.cmu.edu

## Abstract

Recent advances in diffusion models have demonstrated their strong capabilities in generating high-fidelity samples from complex distributions through an iterative refinement process. Despite the empirical success of diffusion models in motion planning and control, the model-free nature of these methods does not leverage readily available model information and limits their generalization to new scenarios beyond the training data (e.g., new robots with different dynamics). In this work, we introduce Model-Based Diffusion (MBD), an optimization approach using the diffusion process to solve trajectory optimization (TO) problems ***without data***. The key idea is to explicitly compute the score function by leveraging the model information in TO problems, which is why we refer to our approach as ***model-based*** diffusion. Moreover, although MBD does not require external data, it can be naturally integrated with data of diverse qualities to steer the diffusion process. We also reveal that MBD has interesting connections to sampling-based optimization. Empirical evaluations show that MBD outperforms state-of-the-art reinforcement learning and sampling-based TO methods in challenging contact-rich tasks. Additionally, MBD's ability to integrate with data enhances its versatility and practical applicability, even with imperfect and infeasible data (e.g., partial-state demonstrations for high-dimensional humanoids), beyond the scope of standard diffusion models. Videos and codes: `https://lecar-lab.github.io/mbd/`

## 1 Introduction

Trajectory optimization (TO) aims to optimize the state and control sequence to minimize a cost function while subject to specified dynamics and constraints. Given nonlinear, non-smooth dynamics and non-convex objectives and constraints, traditional optimization methods like gradient-based methods and interior point methods are less effective in solving TO problems. In response, diffusion models have emerged as a powerful tool for trajectory generation in complex dynamical systems due to their expressiveness and scalability [12, 54, 34, 33, 40, 5].

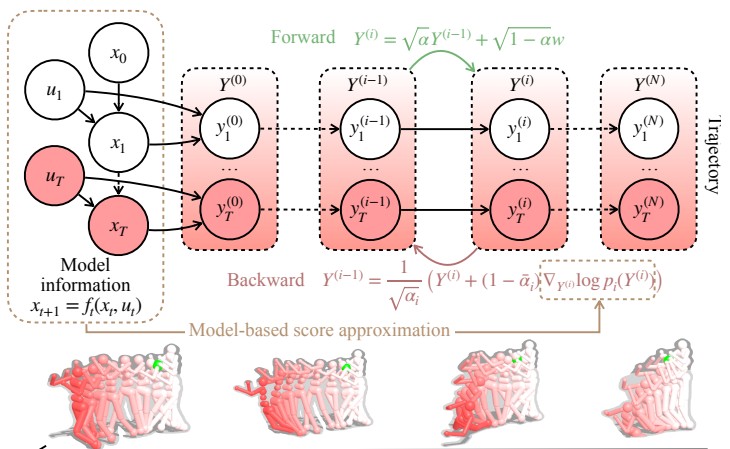

Model-based diffusion in humanoid running tasks

Figure 1: MBD refines the trajectory by leveraging the dynamics model directly without relying on demonstration data.

38th Conference on Neural Information Processing Systems (NeurIPS 2024).

Although diffusion models excel when learning from large-scale, high-dimensional, and high-quality demonstrations, their dependency on such data limits their practicality. For example, after training a manipulation task with a specific robotic arm, the model may struggle to generalize to new tasks with a different arm as the underlying dynamics change. This limitation arises from the model-free nature of existing diffusion-based methods, which do not leverage readily available model information to enhance adaptability. Moreover, existing diffusion-based approaches often require high-quality (in terms of optimality and feasibility) demonstration data, which limits their applications in various scenarios with imperfect data, such as dynamically infeasible trajectories (e.g., generated by high-level planners using simplified models) and partial demonstrations (e.g., lower-body-only demonstrations for a high-dimensional humanoid).

Fortunately, unlike diffusion model's applications in vision or language where data is from unknown distributions (e.g., internet-scale image data), in trajectory optimization, we often know the distribution of desired trajectories, which is described by the optimization objectives, constraints, and the underlying dynamics model, although such a distribution is intractable to directly sample from. Diffusion models offer a tantalizing new perspective, by iteratively refining samples from isotropic Gaussians to meaningful desired distributions in manageable steps, rather than directly learning the complex desired distribution. Inspired by this, we propose Model-Based Diffusion (MBD) that utilizes model information to approximate the gradient of the log probability density function (a.k.a. score function) and uses it to iteratively refine sampled trajectories to solve TO problems, as depicted in Fig. 1. This model-centric strategy allows for the generation of dynamically feasible trajectories in a data-free manner, and gradually moves them towards more optimal solutions. Furthermore, by using demonstrations as observations of the target distribution, MBD can be smoothly combined with data of different qualities to steer the diffusion process and enhance its effectiveness. Particularly, we merge the demonstration data into the sampling process by evaluating their likelihoods with the model and use them to improve the estimation of the score function. Our contributions are threefold:

- We introduce the Model-Based Diffusion (MBD) framework for trajectory optimization, utilizing the dynamics model to estimate the score function. This enables an effective trajectory planner given non-smooth dynamics and non-convex objectives, such as contact-rich manipulation tasks or high-dimensional humanoids.

- Our analysis and empirical evaluations demonstrate that MBD matches, and often exceeds, the performance of existing reinforcement learning and sampling-based TO methods. In particular, MBD outperforms PPO by 59% in various tasks within tens of seconds of diffusing.

- We demonstrate MBD's flexibility in utilizing diverse imperfect data to steer the diffusion process and further enhance the performance. Specifically, the resulting whole-body humanoid trajectory from MBD is more natural by utilizing the lower-body-state-only human motion data. Similarly, MBD can effectively address long-horizon sparse-reward Umaze navigation tasks by leveraging infeasible demonstrations generated by an RRT planner with simplified dynamics.

## 2  Related Work

**Diffusion Models.** Diffusion models have been widely adopted as generative models for high-dimensional data, such as image [51], audio [13], and text [8] through iterative refinement processes [50, 28]. The backward process can be viewed as gradient prediction [52] or score matching [53], which learns the score function to move samples towards the data distribution. We deliver new methods to perform the backward diffusion process using the available model information.

**Sampling-based Optimization.** Optimization involving black-box functions is widely applied across various fields, including hyperparameter tuning and experimental design [49, 27]. Evolutionary algorithms like CMA-ES are often used to tackle black-box optimization problems, dynamically modifying the covariance matrix to produce new samples [24]. Such problems can also be efficiently addressed within the Bayesian optimization framework [50, 19], which offers greater efficiency. Nonetheless, traditional BO algorithms are generally restricted to low-dimensional problems.

**Trajectory Optimization.** Traditionally, trajectory optimization (TO) is solved using gradient-based optimization, which faces challenges such as non-convex problem structures, nonlinear or discontinuous dynamics, and high-dimensional state and control action spaces. As two equivalent formulations, direct methods [25] and shooting-based methods [29] are commonly used to solve TO

problems, where gradient-based optimizers such as Augmented Lagrangian [32], Interior Point [36], and Sequential Quadratic Programming [3, 48] are employed. To leverage the parallelism of modern hardware and improve global convergence properties, sampling-based methods like Cross-Entropy Motion Planning (CEM) [37] and Model Predictive Path Integral (MPPI) [58, 62] have been proposed to solve TO by sampling from target distributions. To solve stochastic optimal control problems, trajectory optimization has also been framed as an inference problem in a probabilistic graphical model, where system dynamics defines the graph structure [35, 39]. This perspective extends methods such as iLQG by integrating approximate inference techniques to improve trajectory optimization [55]. The connection between diffusion and optimal control has been explored in [10], which motivates us to use diffusion models as solvers for trajectory optimization.

**Diffusion for Planning.** Diffusion-based planners have been used to perform human motion generation [12, 54] and multi-agent motion prediction [34]. Diffusion models are capable of generating complete trajectories by folding both dynamics and optimization processes into a single framework, thus mitigating compounding errors and allowing flexible conditioning [33, 40, 5]. In addition, they have been adeptly applied to policy generation, enhancing the capability to capture multimodal demonstration data in high-dimensional spaces for long-horizon tasks [46, 15]. These works assume no access to the underlying dynamics, limiting the generalization to new environments. To enforce dynamics constraints, SafeDiffuser [60] integrates control barrier functions into the learned diffusion process, while Diffusion-CCSP [61] composes the learned geometric and physical conditions to guarantee constraint compliance. Our approach uses diffusion models directly as solvers, rather than simply distilling solutions from demonstrations.

**Langevin-based Markov Chain Monte Carlo for Global Optimization.** Gradient-based sampling algorithms have been widely used in global optimization, where the energy function $J$ is optimized by sampling from the Boltzmann distribution $p \propto \exp(-\frac{J}{\lambda})$ [57, 43]. By annealing the temperature $\lambda$ to zero, the sampling process converges to the global minimum of the energy function $J$ [31]. The convergence of Langevin-based MCMC methods has been well studied in both continuous and discretized settings [16, 21], showing that the distribution will converge in probability to the target distribution with certain decreasing schedule of the step size and temperature $\lambda$. In practice, the most common Langevin-based MCMC methods are unadjusted Langevin Monte Carlo (ULMC) [17] and Underdamped Langevin Monte Carlo (UdLMC) [14], with convergence rates of $O(\frac{1}{\epsilon^2} \log(\frac{1}{\epsilon}))$ and $O(\frac{1}{\epsilon})$ given strongly convex and smooth energy functions, respectively. Recently, Langevin-based MCMC methods have been integrated into diffusion processes to improve global convergence and sampling efficiency [30, 6], where the score function is estimated by Monte Carlo to accelerate and stabilize the diffusion process. Our work differs from these methods in that we aim to sample from the high-probability region of the target distribution without assuming access to the gradient of the energy function, and without assuming the energy function is smooth or convex.

## 3   Problem Statement and Background

**Notations**: We use lower (upper) scripts to specify the time (diffusion) step, e.g., $x_t, u_t, y_t$ represent the state, control and state-control pair at time $t$, and $Y^{(i)}$ represents the diffusion state at step $i$.

This paper focuses on a class of trajectory optimization (TO) problems whose objective is to find the sequences $\{x_t\}$ and $\{u_t\}$ that minimize the cost function $J(x_{1:T}; u_{1:T})$ subject to the dynamics and constraints. The optimization problem [1] can be formulated as follows:

$$\min_{x_{1:T}, u_{1:T}} J(x_{1:T}; u_{1:T}) = l_T(x_T) + \sum_{t=0}^{T-1} l_t(x_t, u_t) \tag{1a}$$

$$\text{s.t.} \quad x_0 = x_{\text{init}} \tag{1b}$$

$$x_{t+1} = f_t(x_t, u_t), \quad \forall t = 0, 1, \dots, T-1, \tag{1c}$$

$$g_t(x_t, u_t) \le 0, \quad \forall t = 0, 1, \dots, T-1.3 \tag{1d}$$

where $x_t \in \mathbb{R}^{n_x}$ and $u_t \in \mathbb{R}^{n_u}$ are the state and control at time $t$, $f_t : \mathbb{R}^{n_x} \times \mathbb{R}^{n_u} \to \mathbb{R}^{n_x}$ represents the dynamics, $g_t : \mathbb{R}^{n_x} \times \mathbb{R}^{n_u} \to \mathbb{R}^{n_g}$ are the constraint functions and $l_t : \mathbb{R}^{n_x} \times \mathbb{R}^{n_u} \to \mathbb{R}$ are the stage

---

[1]We assume deterministic dynamics for simplicity to sample the dynamically feasible trajectory. The extension to stochastic dynamics is straightforward.

costs. We use $Y = [x_{1:T}; u_{1:T}]$ to denote all decision variables. Traditionally, TO is solved using nonlinear programming, which faces challenges such as non-convex problem structures, nonlinear or discontinuous dynamics, and high-dimensional state and control action spaces. Recently, there has been a growing interest in bypassing these challenges by directly generating samples from the optimal trajectory distribution using diffusion models trained on optimal demonstration data [12, 40, 46, 61].

To use diffusion for TO, (1) is first transformed into a sampling problem. The target distribution $p_0(Y^{(0)})$ is proportional to dynamical feasibility $p_d(Y) \propto \prod_{t=1}^{T} \mathbf{1}(x_t = f_{t-1}(x_{t-1}, u_{t-1}))$, optimality $p_J(Y) \propto \exp\left(-\frac{J(Y)}{\lambda}\right)$ and the constraints $p_g(Y) \propto \prod_{t=1}^{T} \mathbf{1}(g_t(x_t, u_t) \le 0)$, i.e.,

$$p_0(Y) \propto p_d(Y)p_J(Y)p_g(Y) \tag{2}$$

Obtaining the solution $Y^*$ from the TO problem in Eq. (1) is equivalent to sampling from Eq. (2) given a low temperature $\lambda \to 0$. In fact, in Appendix A.2, we prove that the distribution of $J(Y)$ with $Y \sim p_0(\cdot)$ converges in probability to the optimal value $J^*$ as $\lambda \to 0$, under mild assumptions. However, it is generally difficult to directly sample from the high-dimensional and sparse target distribution $p_0(\cdot)$. To address this issue, the diffusion process iteratively refines the samples following a backward process, which reverses a predefined forward process as shown in Fig. 1. The forward process corrupts the original distribution $p_0(\cdot)$ to an isotropic Gaussian $p_N(\cdot)$ by incrementally adding small noise to it and scaling it down by $\sqrt{\alpha_i}$ to maintain an invariant noise covariance (see Fig. 2(b) for an example). Mathematically, this means we iteratively obtain $Y^{(i)} \sim p_i(\cdot)$ with $p_{i|i-1}(\cdot|Y^{(i-1)}) \sim \mathcal{N}(\sqrt{\alpha_i}Y^{(i-1)}, (1-\alpha_i)I)$. Because the noise at each time step is independent, the conditional distribution of $Y^{(i)}|Y^{(i-1)}$ also leads to that of $Y^{(i)}|Y^{(0)}$:

$$p_{i|0}(\cdot|Y^{(0)}) \sim \mathcal{N}(\sqrt{\bar{\alpha}_i}Y^{(0)}, (1-\bar{\alpha}_i)I), \quad \bar{\alpha}_i = \prod_{k=1}^{i} \alpha_k. \tag{3}$$

The backward process $p_{i-1|i}(Y^{(i-1)}|Y^{(i)})$ is the reverse of the forward process $p_{i|i-1}(Y^{(i)}|Y^{(i-1)})$, which removes the noise from the corrupted distribution $p_N(\cdot)$ to obtain the target distribution $p_0(\cdot)$. The target distribution $p_0(\cdot)$ in the diffusion process is given by:

$$p_{i-1}(Y^{(i-1)}) = \int p_{i-1|i}(Y^{(i-1)}|Y^{(i)})p_i(Y^{(i)}) \, dY^{(i)}, \tag{4}$$

$$p_0(Y^{(0)}) = \int p_N(Y^{(N)}) \prod_{i=N}^{1} p_{i-1|i}(Y^{(i-1)}|Y^{(i)}) dY^{(1:N)} \tag{5}$$

Standard diffusion models [33, 40, 61], which we refer to as Model-Free Diffusion (MFD), solve the backward process by learning score function merely from data. In contrast, we propose leveraging the dynamics model to estimate the score to improve the generalizability of the model and allow a natural integration with diverse quality data.

## 4 Model-Based Diffusion

In this section, we formally introduce our MBD algorithm that leverages model information to perform backward process. To streamline the discussion, in Section 4.1, we first present MBD with Monte Carlo score ascent to solve simplified and generic unconstrained optimization problems. In Section 4.2, we extend MBD to the constrained optimization setting to solve the TO problem given complex dynamics and constraints. Lastly, in Section 4.3, we augment the MBD algorithm with demonstrations to improve sample quality and steer the diffusion process.

## 4.1 Model-based Diffusion as Multi-stage Optimization

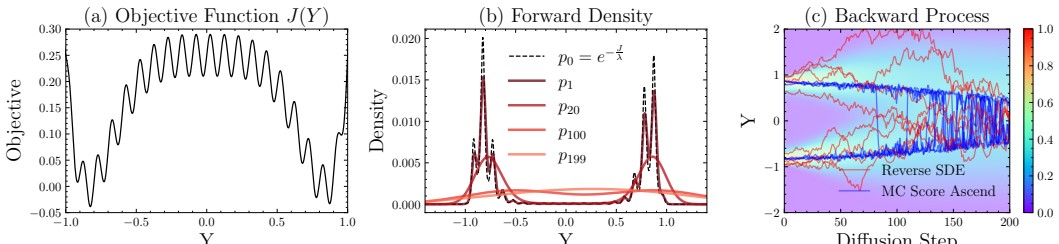

Figure 2: Reverse SDE vs. Monte Carlo score ascent (MCSA) on a synthetic highly non-convex objective function. (a) Synthesized objective function with multiple local minima. (b) The intermediate stage density $p_i(\cdot)$, where peaked $p_0(\cdot)$ is iteratively corrupted to a Gaussian $p_N(\cdot)$. (c) Reverse SDE vs. MCSA: Background colors represent the density of $p_i(\cdot)$ at different stages. MCSA converges faster due to larger step size and lower sampling noise while still capturing the multimodality.

We consider the reverse process for an unconstrained optimization problem $\min_Y J(Y)$, where the target distribution is $p_0(Y^{(0)}) \propto \exp\left(-\frac{J(Y^{(0)})}{\lambda}\right)$. In our MBD framework, "model" implies that we can evaluate $J(Y^{(0)})$ for arbitrary $Y^{(0)}$, enabling us to compute the target distribution up to the normalizing constant.

MBD uses Monte Carlo score ascent instead of the commonly adopted reverse SDE approach in MFD. Specifically, when denoising from $i$ to $i-1$, MBD performs one step of gradient ascent on $\log p_i(Y^{(i)})$ and then scales the sample by the factor $\frac{1}{\sqrt{\alpha_i}}$ as defined in the forward process:

$$Y^{(i-1)} = \frac{1}{\sqrt{\alpha_i}}\left(Y^{(i)} + (1 - \bar{\alpha}_i)\nabla_{Y^{(i)}} \log p_i(Y^{(i)})\right) \tag{6}$$

Critically, with the model-informed $p_0(Y^{(0)})$, we can estimate the score function $\nabla_{Y^{(i)}} \log p_i(Y^{(i)})$ by connecting $p_i(Y^{(i)})$ to $p_0(Y^{(0)})$ via Bayes' rule:

$$\nabla_{Y^{(i)}} \log p_i(Y^{(i)}) = \frac{\nabla_{Y^{(i)}} \int p_{i|0}(Y^{(i)} \mid Y^{(0)}) p_0(Y^{(0)})\, dY^{(0)}}{\int p_{i|0}(Y^{(i)} \mid Y^{(0)}) p_0(Y^{(0)})\, dY^{(0)}} = \frac{\int \nabla_{Y^{(i)}} p_{i|0}(Y^{(i)} \mid Y^{(0)}) p_0(Y^{(0)})\, dY^{(0)}}{\int p_{i|0}(Y^{(i)} \mid Y^{(0)}) p_0(Y^{(0)})\, dY^{(0)}} \tag{7a}$$

$$= \frac{\int -\frac{Y^{(i)} - \sqrt{\bar{\alpha}_i} Y^{(0)}}{1 - \bar{\alpha}_i} p_{i|0}(Y^{(i)} \mid Y^{(0)}) p_0(Y^{(0)})\, dY^{(0)}}{\int p_{i|0}(Y^{(i)} \mid Y^{(0)}) p_0(Y^{(0)})\, dY^{(0)}} \tag{7b}$$

$$= -\frac{Y^{(i)}}{1 - \bar{\alpha}_i} + \frac{\sqrt{\bar{\alpha}_i}}{1 - \bar{\alpha}_i} \frac{\int Y^{(0)} p_{i|0}(Y^{(i)} \mid Y^{(0)}) p_0(Y^{(0)})\, dY^{(0)}}{\int p_{i|0}(Y^{(i)} \mid Y^{(0)}) p_0(Y^{(0)})\, dY^{(0)}} \tag{7c}$$

Between Eq. (7a) and Eq. (7b), we use the forward Gaussian density in Eq. (3): $p_{i|0}(Y^{(i)} \mid Y^{(0)}) \propto \exp\left(-\frac{1}{2}\frac{\left(Y^{(i)} - \sqrt{\bar{\alpha}_i} Y^{(0)}\right)^\top \left(Y^{(i)} - \sqrt{\bar{\alpha}_i} Y^{(0)}\right)}{1 - \bar{\alpha}_i}\right)$. Its log-likelihood gradient is $\nabla_{Y^{(i)}} p_{i|0}(Y^{(i)} \mid Y^{(0)}) = -\frac{1}{1 - \bar{\alpha}_i}(Y^{(i)} - \sqrt{\bar{\alpha}_i} Y^{(0)}) p_{i|0}(Y^{(i)} \mid Y^{(0)})$. Given $Y^{(0)}$ as the integration variable in Eq. (7c), $p_{i|0}(Y^{(i)} \mid Y^{(0)})$ is evaluated as a function of $Y^{(0)}$ parameterized by $Y^{(i)}$. Based on that, we define the function $\phi_i(Y^{(0)})$ as:

$$\phi_i(Y^{(0)}) \propto p_{i|0}(Y^{(i)} \mid Y^{(0)}) \propto \exp\left(-\frac{1}{2}\frac{\left(Y^{(0)} - \frac{Y^{(i)}}{\sqrt{\bar{\alpha}_i}}\right)^\top \left(Y^{(0)} - \frac{Y^{(i)}}{\sqrt{\bar{\alpha}_i}}\right)}{\frac{1 - \bar{\alpha}_i}{\bar{\alpha}_i}}\right) \propto \mathcal{N}\left(\frac{Y^{(i)}}{\sqrt{\bar{\alpha}_i}}, \frac{I}{\bar{\alpha}_i} - I\right) \tag{8}$$

This finding enables the Monte-Carlo estimation for computing the score function. We collect a batch of samples from $\phi_i(\cdot)$ which we denote as $\mathcal{Y}^{(i)}$ and approximate the score as:

$$\nabla_{Y^{(i)}} \log p_i(Y^{(i)}) = -\frac{Y^{(i)}}{1-\bar{\alpha}_i} + \frac{\sqrt{\bar{\alpha}_i}}{1-\bar{\alpha}_i} \frac{\int Y^{(0)} \phi_i(Y^{(0)}) p_0(Y^{(0)}) \, dY^{(0)}}{\int \phi_i(Y^{(0)}) p_0(Y^{(0)}) \, dY^{(0)}} \tag{9a}$$

$$\approx -\frac{Y^{(i)}}{1-\bar{\alpha}_i} + \frac{\sqrt{\bar{\alpha}_i}}{1-\bar{\alpha}_i} \underbrace{\frac{\sum_{Y^{(0)} \in \mathcal{Y}^{(i)}} Y^{(0)} p_0(Y^{(0)})}{\sum_{Y^{(0)} \in \mathcal{Y}^{(i)}} p_0(Y^{(0)})}}_{\text{Monte Carlo Approximation}} := -\frac{Y^{(i)}}{1-\bar{\alpha}_i} + \frac{\sqrt{\bar{\alpha}_i}}{1-\bar{\alpha}_i} \bar{Y}^{(0)}(\mathcal{Y}^{(i)}) \tag{9b}$$

| Aspect | Model-Based Diffusion (MBD) | Model-Free Diffusion (MFD) |
|---|---|---|
| Target Distribution | Known (Eq. (2)), but hard to sample | Unknown, but have data |
| Objective | Sample $Y^{(0)}$ from high-likelihood region of $p_0(\cdot)$ | Sample $Y^{(0)} \sim p_0(\cdot)$ |
| Score Approximation | Estimated using the model (Eq. (9a)). Can be augmented with demonstrations (Eqs. (11) and (13)) | Learned from data |
| Backward Process | Perform Monte Carlo score ascent (Eq. (6)) to move samples towards most-likely states | Run reverse SDE to preserve sample diversity |

Table 1: Comparison of Model-Based Diffusion (MBD) and Model-Free Diffusion (MFD)

**Comparison between MFD and MBD.** Table 1 highlights the key differences between MBD and MFD, which originate from two assumptions made in MBD: (a) a known target distribution $p_0(Y^{(0)})$ given the model; (b) the objective of sampling $Y^{(0)}$ from the high-likelihood region of $p_0(Y^{(0)})$ to minimize the cost function. For (a), MBD leverages $p_0$ to estimate the score following Eq. (9a), whereas MFD learns that from the data. For (b), MBD runs Monte Carlo score ascent in Eq. (6) to quickly move the samples to the high-density region as depicted in Fig. 2(c), while MFD runs reverse SDE $Y^{(i-1)} = \frac{1}{\sqrt{\alpha_i}}\left(Y^{(i)} + \frac{1-\alpha_i}{2}\nabla_{Y^{(i)}} \log p_i(Y^{(i)})\right) + \sqrt{1-\alpha_i}\mathbf{z}_i$, where $\mathbf{z}_i$ is Gaussian noise, to maintain the sample diversity. Given low temperature $\lambda$, it can be shown that $\nabla \log p_i(Y^{(i)}) \approx -\frac{1}{(1-\bar{\alpha}_i)}(Y^{(i)} - \arg\max p_i(\cdot))^2$, i.e., the function $\log p_i(Y^{(i)})$ is $\frac{1}{(1-\bar{\alpha}_i)}$-smooth. Therefore, choosing the step size $(1-\bar{\alpha}_i)$ in Eq. (6) is considered optimal, as for $L$-smooth functions, $O(\frac{1}{L})$ is the step size that achieves the fastest convergence [64].

**How diffusion helps?** The diffusion process plays an important role in helping Monte Carlo score ascent overcome the local minimum issue in highly non-convex optimization problems, as shown in Fig. 2(a). Compared with optimizing a highly non-convex objective, Monte Carlo score ascent is applied to the intermediate distribution $p_i(\cdot) = \int p_0(Y^{(0)})p_{i|0}(\cdot)dY^{(0)}$, which is made concave by convoluting $p_0(\cdot)$ with a Gaussian distribution $p_{i|0}(\cdot)$, as shown in Fig. 2(b). Starting from the strongly concave Gaussian distribution $p_N \sim \mathcal{N}(\mathbf{0}, I)$ with scale $\bar{\alpha}_N \to 0$, the density is easy to sample. The covariance of the sampling density $\Sigma_{\phi_i} = (\frac{1}{\bar{\alpha}_i} - 1)I$ is large when $i = N$, implying that we are searching a wide space for global minima. In the less-noised stage, the intermediate distribution $p_i(\cdot)$ is more peaked and closer to the target distribution $p_0(\cdot)$, and $\bar{\alpha}_i \to 1$ produces a smaller sampling covariance $\Sigma_{\phi_i}$ to perform a local search. By iteratively running gradient ascent on the smoothed distribution, MBD can effectively optimize a highly non-convex objective function as presented in Fig. 2. The MBD algorithm is formally depicted in Algorithm 1.

**Connection with Sampling-based Optimization.** When diffusion step is set to $N = 1$, MBD effectively reduces to the Cross-Entropy Method (CEM) [11] for optimization. To see this, we can plug the estimated score Eq. (9b) into the Monte Carlo score ascent Eq. (6) and set $N = 1$: $Y^{(0)} = \frac{\bar{\alpha}_1}{\alpha_1}\bar{Y}^{(0)}(\mathcal{Y}^{(1)}) = \bar{Y}^{(0)}(\mathcal{Y}^{(1)}) = \frac{\sum_{Y^{(0)} \in \mathcal{Y}^{(1)}} Y^{(0)} w(Y^{(0)})}{\sum_{Y^{(0)} \in \mathcal{Y}^{(1)}} w(Y^{(0)})}$ where $w(Y^{(0)}) = p_0(Y^{(0)}) \propto \exp(-\frac{J(Y^{(0)})}{\lambda})$ and $\mathcal{Y}^{(1)} \sim \mathcal{N}(\frac{Y^{(1)}}{\alpha_0}, (\frac{1}{\alpha_0} - 1)I)$. This precisely mirrors the update mechanism in CEM, which aims to optimize the objective function $f_{\text{CEM}}(Y^{(0)}) = J(Y^{(0)})$ and determine the sampling covariance $\Sigma_{\text{CEM}} = (\frac{1}{\alpha_0} - 1)I$, thus linking the sampling strategy of CEM with the $\alpha$ schedule in MBD. The advances that distinguish MBD from CEM-like methods are (1) the careful

---
[2]See more elaborations in Appendix A.2

---

**Algorithm 1** Model-based Diffusion for Generic Optimization

---
1: **Input:** $Y^{(N)} \sim \mathcal{N}(\mathbf{0}, I)$
2: **for** $i = N$ to 1 **do**
3:    Sample $\mathcal{Y}^{(i)} \sim \mathcal{N}(\frac{Y^{(i)}}{\sqrt{\bar{\alpha}_{i-1}}}, (\frac{1}{\bar{\alpha}_{i-1}} - 1)I)$
4:    Calculate Eq. (9b) $\bar{Y}^{(0)}(\mathcal{Y}^{(i)}) = \frac{\Sigma_{Y^{(0)} \in \mathcal{Y}^{(i)}} Y^{(0)} p_0(Y^{(0)})}{\Sigma_{Y^{(0)} \in \mathcal{Y}^{(i)}} p_0(Y^{(0)})}$
5:    Estimate the score Eq. (9a): $\nabla_{Y^{(i)}} \log p_i(Y^{(i)}) \approx -\frac{Y^{(i)}}{1 - \bar{\alpha}_i} + \frac{\sqrt{\bar{\alpha}_i}}{1 - \bar{\alpha}_i} \bar{Y}^{(0)}(\mathcal{Y}^{(i)})$
6:    Monte Carlo score ascent Eq. (6): $Y^{(i-1)} \leftarrow \frac{1}{\sqrt{\bar{\alpha}_i}} (Y^{(i)} + (1 - \bar{\alpha}_i) \nabla_{Y^{(i)}} \log p_i(Y^{(i)}))$
7: **end for**

---

scheduling of $\alpha$ and (2) the intermediate refinements on $p_i$, both following the forward diffusion process. This allows MBD to optimize for smoothed functions in the early stage and gradually refine the solution for the original objective. On the contrary, CEM's solution could either be biased given a large $\Sigma_{\text{CEM}}$ which overly smoothes the distribution as in $p_{20}, p_{100}$ of Fig. 2(b), or stuck in local minima with a small $\Sigma_{\text{CEM}}$ as in $p_1$ of Fig. 2(b) where the distribution is highly non-concave.

### 4.2 Model-based Diffusion for Trajectory Optimization

For TO, we have to accommodate the constraints in Eq. (1) which change the target distribution to $p_0(Y^{(0)}) \propto p_d(Y^{(0)}) p_J(Y^{(0)}) p_g(Y^{(0)})$. Given that $p_d(Y^{(0)})$ is a Dirac delta function that assigns non-zero probability only to dynamically feasible trajectories, sampling from $\phi_i(Y^{(0)})$ could result in low densities. To enhance sampling efficiency, we collect a batch of dynamically feasible samples $\mathcal{Y}_d^{(i)}$ from the distribution $\phi_i(Y^{(0)}) p_d(Y^{(0)})$ with model information. Proceeding from Eq. (9a), and incorporating $p_0(Y^{(0)}) \propto p_d(Y^{(0)}) p_J(Y^{(0)}) p_g(Y^{(0)})$, we show the score function is:

$$\nabla_{Y^{(i)}} \log p_i(Y^{(i)}) = -\frac{Y^{(i)}}{1 - \bar{\alpha}_i} + \frac{\sqrt{\bar{\alpha}_i}}{1 - \bar{\alpha}_i} \frac{\int Y^{(0)} \phi_i(Y^{(0)}) p_d(Y^{(0)}) p_g(Y^{(0)}) p_J(Y^{(0)}) \, dY^{(0)}}{\int \phi_i(Y^{(0)}) p_d(Y^{(0)}) p_g(Y^{(0)}) p_J(Y^{(0)}) \, dY^{(0)}} \quad (10a)$$

$$\approx -\frac{Y^{(i)}}{1 - \bar{\alpha}_i} + \frac{\sqrt{\bar{\alpha}_i}}{1 - \bar{\alpha}_i} \frac{\Sigma_{Y^{(0)} \in \mathcal{Y}_d^{(i)}} Y^{(0)} p_J(Y^{(0)}) p_g(Y^{(0)})}{\Sigma_{Y^{(0)} \in \mathcal{Y}_d^{(i)}} p_J(Y^{(0)}) p_g(Y^{(0)})} \quad (10b)$$

$$= -\frac{Y^{(i)}}{1 - \bar{\alpha}_i} + \frac{\sqrt{\bar{\alpha}_i}}{1 - \bar{\alpha}_i} \bar{Y}^{(0)}, \quad (10c)$$

$$\text{where} \quad \bar{Y}^{(0)} = \frac{\Sigma_{Y^{(0)} \in \mathcal{Y}_d^{(i)}} Y^{(0)} w(Y^{(0)})}{\Sigma_{Y^{(0)} \in \mathcal{Y}_d^{(i)}} w(Y^{(0)})}, \quad w(Y^{(0)}) = p_J(Y^{(0)}) p_g(Y^{(0)}) \quad (10d)$$

The model plays a crucial role in score estimation by transforming infeasible samples $\mathcal{Y}^{(i)}$ from Line 3 in Algorithm 2 into feasible ones $\mathcal{Y}_d^{(i)}$. The conversion is achieved by putting the control part $U = u_{1:T}$ of $Y^{(0)} = [x_{1:T}; u_{1:T}]$ into the dynamics Eq. (1c) recursively to get the dynamically feasible samples $Y_d^{(0)}$ (Line 4), which shares the same idea with the shooting method [29] in TO. MBD then evaluates the weight of each sample with $p_g(Y^{(0)}) p_J(Y^{(0)})$ in Line 5. One common limitation of shooting methods is that they could be inefficient for long-horizon tasks due to the combinatorial explosion of the constrained space $p_g(Y) \propto \prod_{t=1}^{T} \mathbf{1}(g_t(x_t, u_t) \leq 0)$, leading to low constraint satisfaction rates. To address this issue, we will introduce demonstration-augmented MBD in Section 4.3 to guide the sampling process from the state space to improve sample quality.

### 4.3 Model-based Diffusion with Demonstration

With the ability to leverage model information, MBD can also be seamlessly integrated with various types of data, including imperfect or partial-state demonstrations by modeling them as noisy observations of the desired trajectory $p(Y_{\text{demo}} \mid Y^{(0)}) \sim \mathcal{N}(Y^{(0)}, \sigma^2 I)$. Given suboptimal demonstrations, sampling from the posterior $p(Y^{(0)} \mid Y_{\text{demo}}) \propto p_0(Y^{(0)}) p(Y_{\text{demo}} \mid Y^{(0)})$ could lead to poor solutions as the demonstration likelihood $p(Y_{\text{demo}} \mid Y^{(0)})$ could dominate the model-based distribution $p_0(Y^{(0)}) \propto p_d(Y^{(0)}) p_J(Y^{(0)}) p_g(Y^{(0)})$ and mislead the sampling pro-

---

**Algorithm 2** Model-based Diffusion for Trajectory Optimization

---
1: **Input:** $Y^{(N)} \sim \mathcal{N}(\mathbf{0}, I)$
2: **for** $i = N$ to 1 **do**
3:     Sample $\mathcal{Y}^{(i)} \sim \mathcal{N}(\frac{Y^{(i)}}{\sqrt{\bar{\alpha}_{i-1}}}, (\frac{1}{\bar{\alpha}_{i-1}} - 1)I)$
4:     Get dynamically feasible samples: $\mathcal{Y}_d^{(i)} \leftarrow \text{rollout}(\mathcal{Y}^{(i)})$
5:     Calculate $\bar{Y}^{(0)}$ with Eq. (10d) (model only) or Eq. (13) (model + demonstration)
6:     Estimate the score Eq. (10c): $\nabla_{Y^{(i)}} \log p_i(Y^{(i)}) \approx -\frac{Y^{(i)}}{1-\bar{\alpha}_i} + \frac{\sqrt{\bar{\alpha}_i}}{1-\bar{\alpha}_i} \bar{Y}^{(0)}$
7:     Monte Carlo score ascent Eq. (6): $Y^{(i-1)} \leftarrow \frac{1}{\sqrt{\bar{\alpha}_i}} \left( Y^{(i)} + (1 - \bar{\alpha}_i) \nabla_{Y^{(i)}} \log p_i(Y^{(i)}) \right)$
8: **end for**

---

cess. Rather, we assess $Y^{(0)}$ using $p(Y_{\text{demo}} \mid Y^{(0)})$, employing a similar technique to interchange the distribution's parameter with the random variable, as demonstrated in Eq. (8), to establish $p_{\text{demo}}(Y^{(0)}) \propto p(Y_{\text{demo}} \mid Y^{(0)}) \propto \mathcal{N}(Y^{(0)} \mid Y_{\text{demo}}, \sigma^2 I)$.

To accommodate demonstrations of varying qualities, instead of fixing target to $p_0(Y^{(0)}) p(Y_{\text{demo}} \mid Y^{(0)})$, we propose seperating the $p_0(Y^{(0)})$ from $p_{\text{demo}}(Y^{(0)})$ to form a new target distribution[3]:

$$p_0'(Y^{(0)}) \propto (1 - \eta) p_d(Y^{(0)}) p_J(Y^{(0)}) p_g(Y^{(0)}) + \eta p_{\text{demo}}(Y^{(0)}) p_J(Y_{\text{demo}}) p_g(Y_{\text{demo}}) \quad (11)$$

where $\eta$ is a constant to balance the model and the demonstration. Here, we have introduced two extra constant terms $p_J(Y_{\text{demo}})$ and $p_g(Y_{\text{demo}})$ to ensure that the demonstration likelihood is properly scaled to match the model likelihood $p_0(Y^{(0)})$. With these preparations, we propose to adaptively determine the significance of the demonstration by choosing $\eta$ as follows:

$$\eta = \begin{cases} 1 & p_d(Y^{(0)}) p_J(Y^{(0)}) p_g(Y^{(0)}) < p_{\text{demo}}(Y^{(0)}) p_J(Y_{\text{demo}}) p_g(Y_{\text{demo}}) \\ 0 & p_d(Y^{(0)}) p_J(Y^{(0)}) p_g(Y^{(0)}) \geq p_{\text{demo}}(Y^{(0)}) p_J(Y_{\text{demo}}) p_g(Y_{\text{demo}}). \end{cases} \quad (12)$$

When samples have a high model-likelihood $p_0$, we ignore the demonstration and sample from the model. Otherwise, we trust the demonstration. With the demonstration-augmented target distribution, we modify the way to calculate $\bar{Y}^{(0)}$ in Eq. (10d) as follows to obtain the score estimate:

$$\bar{Y}^{(0)} = \frac{\sum_{Y^{(0)} \in \mathcal{Y}_d^{(i)}} Y^{(0)} w(Y^{(0)})}{\sum_{Y^{(0)} \in \mathcal{Y}_d^{(i)}} w(Y^{(0)})}, \quad w(Y^{(0)}) = \max \left\{ \begin{array}{c} p_d(Y^{(0)}) p_J(Y^{(0)}) p_g(Y^{(0)}), \\ p_{\text{demo}}(Y^{(0)}) p_J(Y_{\text{demo}}) p_g(Y_{\text{demo}}) \end{array} \right\}. \quad (13)$$

## 5 Experimental Results

The experimental section will focus on demonstrating the capabilities of MBD in: (1) its effectiveness as a zeroth-order solver for high-dimensional, non-convex, and non-smooth trajectory optimization problems, and (2) its flexibility in utilizing dynamically infeasible data to enhance performance and regularize solutions. Our benchmark shows that MBD outperforms PPO by $59\%$ in various control tasks with $10\%$ computational time.

Beyond control problems, in Appendix A.3, we also show that MBD significantly improves sampling efficiency by an average of $23\%$ over leading baselines in high-dimensional (up to 800d) black-box optimization testbeds [23, 18, 56, 42, 41, 44]. We also apply MBD to optimize an MLP network with 28K parameters in a *gradient-free* manner, achieving $86\%$ accuracy within 2s for the MNIST classification task [2], which is comparable to the gradient-based optimizer (SGD with momentum, $93\%$ accuracy). To further extend MBD to closed-loop control, we employ receding horizon strategy to MBD in Appendix A.6 to update control sequence at each timestep, further improving the performance of MBD by $9.6\%$ in terms of reward.

### 5.1 MBD for Planning in Contact-rich Tasks

To test the effectiveness of MBD as a trajectory optimizer for systems involving non-smooth dynamics, we run MBD on both locomotion and manipulation tasks detailed in Appendix A.5.1. The locomotion

---

[3]A comparison between the demonstration-augmented MBD and the vanilla MBD is illustrated in Fig. 6 with detailed breakdowns in Appendix A.4.

| Task | CMA-ES | CEM | MPPI | RL* | MBD |
|------|--------|-----|------|-----|-----|
| Hopper | $1.12 \pm 0.10$ | $0.65 \pm 0.12$ | $0.91 \pm 0.15$ | $1.40 \pm 0.04$ | $\mathbf{1.53 \pm 0.03}$ |
| Half Cheetah | $0.44 \pm 0.10$ | $0.22 \pm 0.15$ | $0.20 \pm 0.14$ | $1.59 \pm 0.05$ | $\mathbf{2.31 \pm 0.19}$ |
| Ant | $1.18 \pm 0.52$ | $0.85 \pm 0.17$ | $0.33 \pm 0.45$ | $3.26 \pm 1.61$ | $\mathbf{3.80 \pm 0.35}$ |
| Walker2D | $0.83 \pm 0.04$ | $1.06 \pm 0.04$ | $0.90 \pm 0.05$ | $1.09 \pm 0.28$ | $\mathbf{2.63 \pm 0.23}$ |
| Humanoid Standup | $0.58 \pm 0.01$ | $0.47 \pm 0.01$ | $0.53 \pm 0.05$ | $0.83 \pm 0.02$ | $\mathbf{0.99 \pm 0.07}$ |
| Humanoid Running | $0.60 \pm 0.11$ | $0.41 \pm 0.16$ | $0.59 \pm 0.14$ | $1.80 \pm 0.03$ | $\mathbf{2.92 \pm 0.26}$ |
| Push T | $0.39 \pm 0.07$ | $0.25 \pm 0.09$ | $-0.13 \pm 0.09$ | $-0.63 \pm 0.16$ | $\mathbf{0.67 \pm 0.10}$ |

Table 2: Reward of different methods on non-continuous tasks. *RL requires offline training and generate a closed-loop policy so it is not an apple-to-apple baseline.

| Task | CMA-ES | CEM | MPPI | RL | MBD |
|------|--------|-----|------|-----|-----|
| Hopper | $29.3\,\mathrm{s}$ | $26.5\,\mathrm{s}$ | $26.4\,\mathrm{s}$ | $17\,\mathrm{m}\,45.63\,\mathrm{s}$ | $26.5\,\mathrm{s}$ |
| Half Cheetah | $29.5\,\mathrm{s}$ | $26.4\,\mathrm{s}$ | $26.7\,\mathrm{s}$ | $4\,\mathrm{m}\,18.8\,\mathrm{s}$ | $26.8\,\mathrm{s}$ |
| Ant | $18.4\,\mathrm{s}$ | $16.1\,\mathrm{s}$ | $16.0\,\mathrm{s}$ | $2\,\mathrm{m}\,46.8\,\mathrm{s}$ | $16.2\,\mathrm{s}$ |
| Walker2D | $37.5\,\mathrm{s}$ | $34.5\,\mathrm{s}$ | $34.7\,\mathrm{s}$ | $5\,\mathrm{m}\,1.5\,\mathrm{s}$ | $34.6\,\mathrm{s}$ |
| Humanoid Standup | $20.8\,\mathrm{s}$ | $17.6\,\mathrm{s}$ | $17.7\,\mathrm{s}$ | $4\,\mathrm{m}\,29\,\mathrm{s}$ | $17.7\,\mathrm{s}$ |
| Humanoid Running | $30.8\,\mathrm{s}$ | $29.7\,\mathrm{s}$ | $29.6\,\mathrm{s}$ | $3\,\mathrm{m}\,34.7\,\mathrm{s}$ | $30.0\,\mathrm{s}$ |
| Push T | $10\,\mathrm{m}\,40.0\,\mathrm{s}$ | $10\,\mathrm{m}\,32.0\,\mathrm{s}$ | $10\,\mathrm{m}\,32.3\,\mathrm{s}$ | $67\,\mathrm{m}\,25.6\,\mathrm{s}$ | $10\,\mathrm{m}\,32.8\,\mathrm{s}$ |

Table 3: Computational time of different methods on non-continuous tasks.

tasks includes hopper, half-cheetah, ant, walker2d, humanoid-standup, and humanoid-running. The selected manipulation task, pushT [15], presents its own challenges due to the complexity introduced by contact dynamics and the long-horizon nature of the task. These tasks are widely considered difficult due to their hybrid nature and high dimensionality.

MBD is compared with the state-of-the-art zeroth-order optimization methods, including CMA-ES [7], CEM [11], and MPPI [59], as well as reinforcement learning (RL) algorithms (e.g., PPO [47] and SAC [22]) on these tasks. Please note RL is only used for performance reference not as there is no existing TO method that can solve such high-dimensional discontinuous tasks as we have shown in the experiments. Model-free RL, especially PPO/SAC, is widely used in such tasks and is considered the SOTA method. The difference between RL and MBD is further discussed in Appendix A.8 and A.6. The RL implementation follows the high-performance parallelized framework from Google Brax [20] elaborated in Appendix A.5.3. For the zeroth-order optimizer, we match the iteration and sample number with the MBD. All the experiments were conducted on a single NVIDIA RTX 4070 Ti GPU. Quantitative metrics including the average step reward and the computational time tested over 50 steps repeated for 8 seeds are reported in Tables 2 and 3. MBD substantially outperforms zeroth-order optimization methods and even outperforms RL in most tasks. Specifically, for the pushT task, MBD achieves a significantly higher reward than the RL algorithm thanks to its iterative refinement process, which effectively explores the full control space while keeping fine-grained control to precisely push the object. Compared with the computationally heavy RL algorithms, MBD only requires one-tenth of time, which is similar to other zeroth-order optimization methods. The optimization process of MBD is visualized in Fig. 3, where the iterative refinement process with the model plays a crucial role in optimizing high-dimensional tasks.

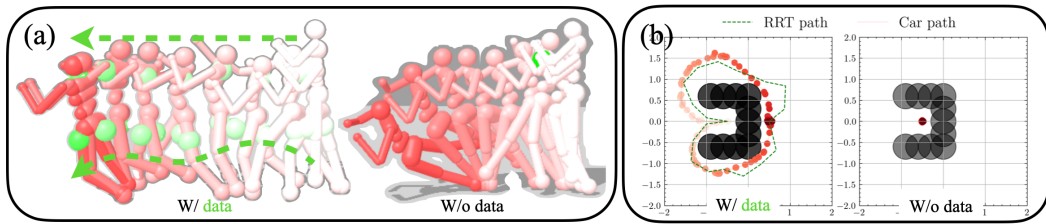

Figure 4: MBD optimized trajectory with data augmentation on the (a) Humanoid Jogging and (b) Car UMaze Navigation tasks. With data augmentation, the trajectory is regularized and refined to achieve the desired objective.

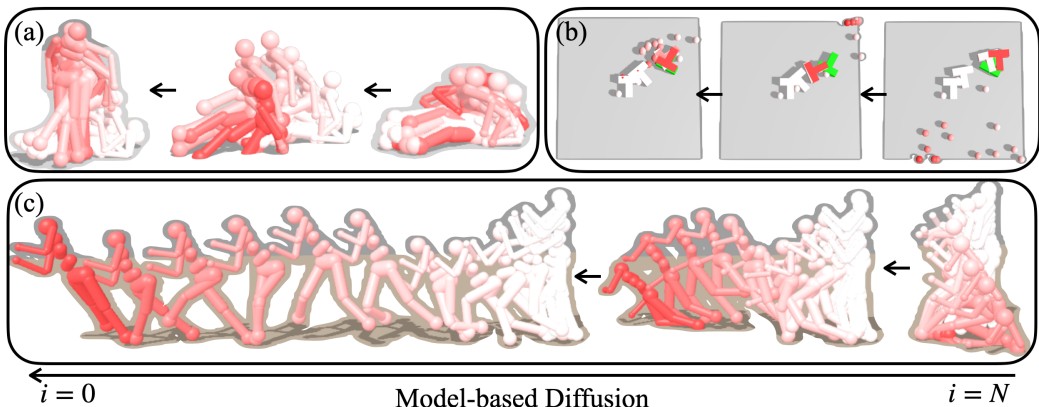

$i = 0$        Model-based Diffusion        $i = N$

Figure 3: Optimization process of MBD on the (a) Humanoid Standup, (b) Push T, and (c) Humanoid Running tasks. The trajectory is iteratively refined to achieve the desired objective in the high-dimensional space with model information.

## 5.2 Data-augmented MBD for Trajectory Optimization

We also evaluate the performance of MBD with data augmentation on the Car UMaze Navigation and Humanoid Jogging tasks to see how partial and dynamically infeasible data can help the exploration of MBD and regularize the solution by steering the diffusion process.

For Car UMaze Navigation, the map blocked by U-shaped obstacles is challenging to explore given a nonlinear dynamics model. Therefore, random shooting has a low chance of reaching the goal region. To sample with loosened dynamical constraints, we augment MBD with data from the RRT [38] algorithm through goal-directed exploration with simplified dynamics. Fig. 4(b) shows the difference between data-augmented MBD and data-free one: the former can refine the infeasible trajectory and further improve it to reach the goal in less time, while the latter struggles to find a feasible solution. The reason is that the infeasible trajectory from RRT serves as a good initialization for MBD, which can be further refined to minimize the cost function with MBD.

For Humanoid Jogging, we aim to regularize the solution for the task with multiple solutions to the desired one with partial state data. Due to the infinite possibilities for humanoid jogging motion, the human motion data provide a good reference to regularize MBD to converge to a more human-like and robust solution instead of an aggressive or unstable one [26, 45]. We use data from the CMU Mocap dataset [1], from which we extract torso, thigh, and shin positions and use them as a partial state reference. Fig. 4(a) demonstrates a more stable motion generated by data-augmented MBD.

## 6 Conclusion and Future Work

This paper introduces Model-Based Diffusion (MBD), a novel diffusion-based trajectory optimization framework that employs a dynamics model to approximate the score function. MBD not only out-performs existing methods in terms of sample efficiency and generalization, but also provides a new perspective on trajectory optimization by leveraging diffusion models as powerful samplers. Future directions involve theoretically understanding its convergence, optimizing the standard Gaussian forward process using the model information, adapting it to online tasks with receding horizon strategies, and exploring advanced sampling and scheduling techniques to further improve performance.

### Acknowledgments

This work was supported by NSF Grant 2154171, NSF CAREER 2339112 and CMU CyLab Seed Funding.

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

# A Appendix / Supplemental Material

## A.1 Notation Table

| Meaning | Symbol |
|---|---|
| state, control at time t | $x_t, u_t$ |
| state control pair at time t | $y_t$ |
| diffused random variable at step i | $Y^{(i)}$ |
| density of diffused distribution at step i | $p_i(\cdot)$ |
| density of diffused r.v. at step i conditioned on step j | $p_{i|j}(\cdot\,|\,\cdot)$ |
| samples collected from proposal distribution at setp i | $\mathcal{Y}^{(i)}$ |
| scale down factor at step i | $\alpha_i$ |
| accumulated scale down factor at step i | $\bar{\alpha}_i$ |
| dynamic feasibility density, optimality density, constraint density | $p_d(\cdot), p_J(\cdot), p_g(\cdot)$ |

## A.2 Convergence of Distribution with Small $\lambda$

We first give the definition of the volume of the sub-level set for cost $J$.

**Definition 1.** *Let $F : \mathcal{R}^d \to \mathcal{R}$ be a measurable function. Define the volume of the sub-level set for a given level $t$ as:*

$$V_F(t) = \int_{\mathcal{R}^d} \chi_{\{Y \in \mathcal{R}^d : F(Y) \le t\}}(Y)\, dY,$$

*where $\chi$ denotes the indicator function.*

The volume function $V_J(t)$ plays a crucial role in linking geometric properties with probabilistic outcomes in optimization and learning algorithms. This function provides a quantitative measure that helps us to understand how changes in parameters like $\lambda$ influence the distribution and concentration of probability mass.

The interplay between geometry and probability, represented by $V_J(t)$, is crucial for evaluating the convergence and stability of algorithms. It provides a significant method for utilizing the PDF of the random variable $\mathbf{Y}$ to constrain the CDF, thereby facilitating convergence in distribution.

**Proposition 2.** *Given the target distribution $\mathbf{Y} \sim p(\cdot)$ with $P(Y) \propto \exp\left(-\frac{J(Y)}{\lambda}\right), Y \in \mathcal{R}^d$, where $J$ is a cost function with $\min_Y J(Y) = 0$ and $Y^* = \arg\min J(Y)$, and assuming that the volume function $V_J(t)$ is bounded by polynomial inequalities:*

$$Poly_l(t) \le V_J(t) \le Poly_u(t),$$

*where $Poly_l(t) = \sum_{k=0}^{M} c_k^l t^{\alpha_k}$ and $Poly_u(t) = \sum_{k=0}^{M} c_k^u t^{\alpha_k}$ are polynomials with coefficients satisfying $c_k^l = 0$ if and only if $c_k^u = 0$. The exponent term satisfies that $\alpha_k \in \mathcal{R}$, and $0 < \alpha_0 < \alpha_1 < \cdots < \alpha_M < \infty$, It follows that:*

$$\lim_{\lambda \to 0} J(\mathbf{Y}) \xrightarrow{p} J(Y^*) = 0.$$

*The cost value $J(Y)$ converges in probability to $J(Y^*)$ as $\lambda \to 0$.*

The condition on the polynomial bounds of $V_J(t)$ is generally not restrictive. For instance, consider $J = \eta_c \|Y - Y_*\|^m$, where $Y^*$ is the optimal point and $\eta_c > 0$ is any constant. In this case, $V_J(t) = Ct^{\frac{d}{m}}$, where $C$ is a constant, meets the constraint in a straightforward way. This condition can be extended beyond this simple scenario, as even if $J$ has multiple modes, it can still adhere to this polynomial constraint.

*Proof.* The convergence in distribution of $\mathbf{Y}$ towards $Y^*$ as $\lambda \to 0$ is established by analyzing the behavior of the probability density function, defined up to a multiplicative constant. Consider the

density $\boldsymbol{Y}_0 \sim p_\lambda(Y)$ approximating $Y^*$ when $\lambda$ approaches zero.

$$P(J(\boldsymbol{Y}) \le t) = \int_{\{J(Y) \le t\}} p(Y) dY, \tag{14a}$$

$$= \int_0^t \int_{\{J(Y)=x\}} p(Y) dY dx, \tag{14b}$$

$$\propto \int_0^t exp(-\frac{x}{\lambda}) \int_{\{J(Y)=x\}} dY dx, \tag{14c}$$

$$= \int_0^t exp(-\frac{x}{\lambda}) \frac{dV_J(x)}{dx} dx, \tag{14d}$$

where Eq. (14c) is valid since $P(Y) \propto \exp\left(-\frac{J(Y)}{\lambda}\right)$ and $J(Y)$ represents the sufficient statistics of the distribution. We can obtain Eq. (14d) by computing the derivative of $V_J(x)$ based on the volume definition as shown in Definition 1.

We denote $J_{\min} = \min_Y J(Y) = 0$ and $J_{\max} = \max_Y J(Y)$ with $J_{\max}$ satisfying $0 \le J_{\max} \le +\infty$. We proceed to analyze Eq. (14d) by performing integration by parts as shown in Eq. (15a).

$$\int_{J_{\min}}^{J_{\max}} \exp\left(-\frac{t}{\lambda}\right) dV_J(t) = \int_{J_{\min}}^{J_{\max}} d\left[\exp\left(-\frac{t}{\lambda}\right) V_J(t)\right] + \frac{1}{\lambda} \exp\left(-\frac{t}{\lambda}\right) V_J(t) dt \tag{15a}$$

$$= \exp\left(-\frac{t}{\lambda}\right) V_J(t)\Big|_{J_{\min}}^{J_{\max}} + \frac{1}{\lambda} \int_{J_{\min}}^{J_{\max}} \exp\left(-\frac{t}{\lambda}\right) V_J(t) dt. \tag{15b}$$

To establish convergence in probability, we need to demonstrate that for any small $\epsilon > 0$ and $\delta > 0$, there exists sufficiently small $\lambda > 0$, such that

$$P(J(\boldsymbol{Y}) < \epsilon) = \frac{\int_0^\epsilon \exp\left(-\frac{t}{\lambda}\right) dV_J(t)}{\int_0^{J_{\max}} \exp\left(-\frac{t}{\lambda}\right) dV_J(t)} \ge 1 - \delta. \tag{16}$$

where the equality is due to Eq. (14d). Setting $\delta' = \frac{1-\delta}{\delta}$, it suffices to show that:

$$\frac{\int_0^\epsilon \exp\left(-\frac{t}{\lambda}\right) dV_J(t)}{\int_\epsilon^{J_{\max}} \exp\left(-\frac{t}{\lambda}\right) dV_J(t)} \ge \delta'. \tag{17}$$

Assuming without loss of generality that $J_{\max} = \infty$, becuase $dV_J(t) \ge 0$, $exp(-\frac{t}{\lambda}) > 0$, we have:

$$\frac{\int_0^\epsilon \exp\left(-\frac{t}{\lambda}\right) dV_J(t)}{\int_\epsilon^{J_{\max}} \exp\left(-\frac{t}{\lambda}\right) dV_J(t)} \ge \frac{\int_0^\epsilon \exp\left(-\frac{t}{\lambda}\right) dV_J(t)}{\int_\epsilon^\infty \exp\left(-\frac{t}{\lambda}\right) dV_J(t)}. \tag{18}$$

This ratio as in Eq. (17) can be expanded using the integral bounds and the polynomial approximations for $V_J(t)$, then it suffices to show that

$$\frac{\int_0^\epsilon \exp\left(-\frac{t}{\lambda}\right) dV_J(t)}{\int_\epsilon^\infty \exp\left(-\frac{t}{\lambda}\right) dV_J(t)} \ge \delta'. \tag{19}$$

By inserting Eq. (15b) into both the numerator and denominator on the LHS of Eq. (19), we obtain

$$\frac{\int_0^\epsilon exp(-\frac{t}{\lambda}) dV_J(t)}{\int_\epsilon^\infty exp(-\frac{t}{\lambda}) dV_J(t)} = \frac{exp(-\frac{\epsilon}{\lambda}) V(\epsilon) + \frac{1}{\lambda}\int_0^\epsilon exp(-\frac{t}{\lambda}) V_J(t) dt}{-exp(-\frac{\epsilon}{\lambda}) V(\epsilon) + \frac{1}{\lambda}\int_\epsilon^\infty exp(-\frac{t}{\lambda}) V_J(t) dt} \tag{20a}$$

$$\ge \frac{\int_0^\epsilon exp(-\frac{t}{\lambda}) V_J(t) dt}{\int_\epsilon^\infty exp(-\frac{t}{\lambda}) V_J(t) dt} \tag{20b}$$

$$\ge \frac{\int_0^\epsilon exp(-\frac{t}{\lambda}) \sum_{k=0}^M c_k^l t^{\alpha_k} dt}{\int_\epsilon^\infty exp(-\frac{t}{\lambda}) \sum_{k=0}^M c_k^u t^{\alpha_k} dt} \tag{20c}$$

To bound the expression in Eq. (20c), we first derive the following integrals by utilizing a change of variables $x = \frac{t}{\lambda}$, which simplifies the expressions:

$$\int_0^\epsilon \exp\left(-\frac{t}{\lambda}\right) t^{\alpha_k}\, dt = \lambda^{k+1} \int_0^{\frac{\epsilon}{\lambda}} \exp(-x) x^{\alpha_k}\, dx, \tag{21a}$$

$$\int_{\frac{\epsilon}{\lambda}}^\infty \exp\left(-\frac{t}{\lambda}\right) t^{\alpha_k}\, dt = \lambda^{k+1} \int_{\frac{\epsilon}{\lambda}}^\infty \exp(-x) x^{\alpha_k}\, dx. \tag{21b}$$

For these transformed integrals, we can observe that $\int_0^\infty \exp(-x) x^{\alpha_k}\, dx = \Gamma(\alpha_k + 1)$, the gamma function, which is well-defined for all non-negative $\alpha_k$. Given that $\delta'$ is a function of $\delta$, by applying the intermediate value theorem and definition of the limit of the integral, we can choose $\epsilon_k$ in such a way that:

$$\int_0^{\epsilon_k} \exp(-x) x^{\alpha_k}\, dx \geq \frac{c_k \delta'}{1 + c_k \delta'} \Gamma(\alpha_k + 1),$$

where $c_k = \frac{c_k^l}{c_k^u}$ denotes the ratio of coefficients in polynomial lower and upper bounds for $V_J(t)$. By selecting $\epsilon_{\max} = \max \epsilon_0, \epsilon_1, \cdots, \epsilon_M$ to be the maximum of all such $\epsilon_k$, ensuring coverage for all polynomial terms up to $M$, we establish that:

$$\frac{\int_0^{\epsilon_{\max}} \exp\left(-\frac{t}{\lambda}\right) c_k^l t^{\alpha_k}\, dt}{\int_{\epsilon_{\max}}^\infty \exp\left(-\frac{t}{\lambda}\right) c_k^u t^{\alpha_k}\, dt} \geq \delta', \quad \text{for all } k = 0, 1, \ldots, M. \tag{22}$$

By ensuring that $\lambda \leq \frac{\epsilon}{\epsilon_{\max}}$, we can conclude:

$$\frac{\int_0^\epsilon \exp\left(-\frac{t}{\lambda}\right) \sum_{k=0}^M c_k^l t^{\alpha_k}\, dt}{\int_\epsilon^\infty \exp\left(-\frac{t}{\lambda}\right) \sum_{k=0}^M c_k^u t^{\alpha_k}\, dt} \geq \delta', \tag{23}$$

Thus, the condition specified in Eq. (16) is satisfied, validating that the distribution of $\mathbf{Y}$ converges in distribution to $Y^*$ as $\lambda$ approaches zero.

$\square$

By adding another mild assumption regarding the landscape of $J$ near the global optimum, we can demonstrate the convergence of the random variable $\mathbf{Y}$ itself, rather than the convergence of $J(\mathbf{Y})$.

**Definition 3.** *We denote the minimum of the complementary set of neighborhood as:*

$$J_\mathcal{B}^*(\delta) = \min_{\|Y - Y^*\| > \delta} J(Y) - J(Y^*).$$

**Proposition 4.** *Given the context and conditions specified in Definitions 1 and 3 and Proposition 2, and given that $J$ has only one golbal minimizer $Y^*$, i.e. there exist small $\delta^*$, that for $\delta \in (0, \delta^*]$, $J_\mathcal{B}^*(\delta)$ is strictly increasing, and $J_\mathcal{B}^*(\delta^*) < \infty$. It follows that:*

$$\lim_{\lambda \to 0} \mathbf{Y} \xrightarrow{p} Y^*.$$

*The random variable $\mathbf{Y}$ converges in probability to $Y^*$ as $\lambda \to 0$.*

*Proof.* In order to prove that $\lim_{\lambda \to 0} \mathbf{Y} \xrightarrow{p} Y^*$. We need to prove that for any sufficient small $\gamma > 0$ and $\delta > 0$, there exists small $\lambda > 0$, such that

$$P(\|Y - Y^*\| \leq \delta) \geq 1 - \gamma \tag{24}$$

From Definition 3 and due to the strict increase of $J_\mathcal{B}^*(\delta)$,

$$\|Y - Y^*\| \leq \delta, \quad \forall Y \in \left\{ Y \in \mathcal{R}^d \mid J(Y) - J(Y^*) < J_\mathcal{B}^*(\delta) \right\}, \tag{25}$$

where $0 < \delta \leq \delta^*$. Because if $\|Y - Y^*\| > \delta$, $J(Y) - J(Y^*) < J_\mathcal{B}^*(\delta) = \min_{\|Y - Y^*\| > \delta} J(Y) - J(Y^*)$ contradicts Definition 3.

Given that $\lim_{\lambda \to 0} J(\boldsymbol{Y}) \xrightarrow{p} J(Y^*)$. and any sufficient small $\epsilon, \gamma > 0$.

$$P(J(Y) - J(Y^*) \le \epsilon) \ge 1 - \gamma \tag{26}$$

Therefore, $\exists \lambda > 0$, such that

$$P(J(Y) - J(Y^*) \le J_{\mathcal{B}}^*(\delta)) \ge 1 - \gamma. \tag{27}$$

And From Eq. (25), we have that

$$P(\|Y - Y^*\| \le \delta) \ge P(J(Y) - J(Y^*) \le J_{\mathcal{B}}^*(\delta)) \ge 1 - \gamma \tag{28}$$

We have that $\boldsymbol{Y}$ converges in probability to $Y^*$ ,i.e, $\lim_{\lambda \to 0} \boldsymbol{Y} \xrightarrow{p} Y^*$.

$\square$

**Proposition 5.** *Given the context and conditions specified in Propositions 2 and 4 and the way we define the forward process as in Eq. (3). The diffused $Y_i$ converge in density to a Gaussian distribution.*

$$\lim_{\lambda \to 0} \boldsymbol{Y}^{(i)} \xrightarrow{d} \mathcal{N}(\sqrt{\bar{\alpha}_i} Y^*, \sqrt{1 - \bar{\alpha}_i} I),$$

*where $\boldsymbol{Y}^{(i)} \sim p_i(\cdot)$ as in Eq. (3).*

Proposition 5 is derived by using Slutsky's theorem on Proposition 4 and offers insight into choosing the stepsize as discussed in Section 4.1.

### A.3 Black-box Optimization with MBD

As a zeroth order optimizer, MBD is capable of addressing both trajectory optimization and broader, high-dimensional unconstrained optimization challenges. Such black-box optimization tasks are universally acknowledged as difficult [7, 63]. We first show superior performance of MBD within this black-box optimization context. In such settings, the Bayesian Optimization technique struggles due to the computational intensity required to develop surrogate models and identify new potential solutions [18]. Alternative black-box optimization strategies [23] are not limited by computational issues but tend to be less efficient because they do not estimate the black-box function as accurately. MBD's effectiveness is evaluated using two well-known highly non-convex black-box optimization benchmarks: Ackley [4] and Rastrigin [9], each tested across three different dimensionalities. Comparisons were made with CMA-ES [23], TuRBO [18], LA-MCTS [56], HesBO [42], Shiwa [41], and BAxUS [44].

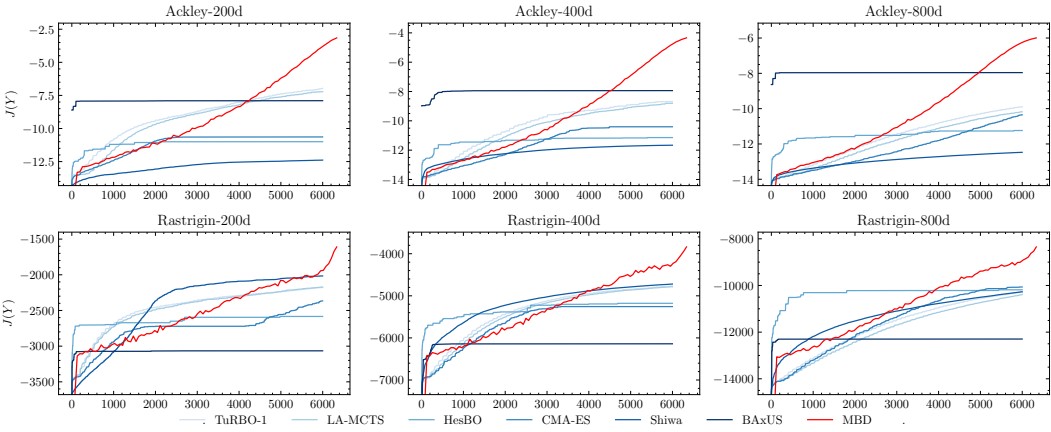

Figure 5: Performance of MBD on high-dimensional black-box optimization benchmarks. MBD outperforms other Gaussian Process-based Bayesian Optimization methods by a clear margin.

Fig. 5 shows the performance of MBD on the Ackley and Rastrigin benchmarks. MBD demonstrates superior performance over other algorithms for several reasons. Firstly, the implementation of a

scheduled forward process that determines the total number of samples consequently boosts sample efficiency. Secondly, the application of Monte Carlo score ascent on various $\log p_i(Y^{(i)})$ facilitates its escape from local optima of varying scales. It is important to acknowledge that comparing computational efficiency may not be entirely fair, given that black-box optimization problems typically involve functions that are costly to evaluate. However, MBD markedly outperforms other Gaussian Process-based Bayesian Optimization approaches, achieving computational time savings of more than twentyfold, similar to the improvements observed with different evolutionary optimization strategies.

Here are the implementation detail for the benchmarks. For the BO benchmarks, the experiments were conducted on an A100 GPU because of the high computational demands of the Gaussian Process Regression Model it incorporates.

**TuRBO**: TuRBO is implemented based on tutorials from Botorch [9].

**LA-MCTS**: LA-MCTS, we refer to authors' reference implementations, and use TuRBO as its local BO solver [56].

**HesBO**: For HesBO, we refer to authors' reference implementations [42]. We transformed default GP component into Gpytorch version for faster inference speed on GPU. We set the embedding dimension to 20 for all tasks

**CMA-ES**: We use pycma[4] to implement CMA-ES, and use default setting except setting population size eqauls to batch size.

**Shiwa**: We use Nevergrad[5] to implement Shiwa, and use default setting to run experiments.

**BAxUS**: We refer to the authors' reference implementations [44].

### A.3.1  MBD for DNN Training without Gradient Information

To further demonstrate the effectiveness of MBD in high-dimensional systems, we apply MBD to optimize an MLP network for MNIST classification [2] without access to the gradient information. MBD achieve $85.5\%$ accuracy with 256 samples within 2s, which is comparable to the performance of the SGD optimizer with momentum ($92.7\%$ accuracy). We use MLP with 2 hidden layers, each with 32 neurons, and ReLU activation function. The input is flattened to 784 dimensions, and the output is a 10-dimensional vector. We use cross-entropy loss as the objective function. The network has $27,562$ parameters in total, which makes sampling-based optimization challenging. MBD can effectively optimize the network with a small number of samples, demonstrating its effectiveness in high-dimensional black-box optimization tasks.

### A.4  MBD with Demonstration Explaination

Data-augmented MBD calculate the score function with demostration as follows:

$$Y^{(i-1)} = \frac{1}{\sqrt{\alpha_i}} \left( Y^{(i)} + (1 - \bar{\alpha}_i) \nabla_{Y^{(i)}} \log p_i(Y^{(i)}) \right) \tag{29}$$

$$\nabla_{Y^{(i)}} \log p_i(Y^{(i)}) = -\frac{Y^{(i)}}{1 - \bar{\alpha}_i} + \frac{\sqrt{\bar{\alpha}_i}}{1 - \bar{\alpha}_i} \bar{Y}^{(0)} \tag{30}$$

$$\text{where} \quad \bar{Y}^{(0)} = \frac{\sum_{Y^{(0)} \in \mathcal{Y}_d^{(i)}} Y^{(0)} w(Y^{(0)})}{\sum_{Y^{(0)} \in \mathcal{Y}_d^{(i)}} w(Y^{(0)})} \tag{31}$$

$$w(Y^{(0)}) = \max \left\{ \begin{array}{c} w_{\text{model}}(Y^{(0)}) = p_d(Y^{(0)}) p_J(Y^{(0)}) p_g(Y^{(0)}), \\ w_{\text{demo}}((Y^{(0)})) = p_{\text{demo}}(Y^{(0)}) p_J(Y_{\text{demo}}) p_g(Y_{\text{demo}}) \end{array} \right\} \tag{32}$$

---

[4]https://github.com/CMA-ES/pycma
[5]https://github.com/facebookresearch/nevergrad

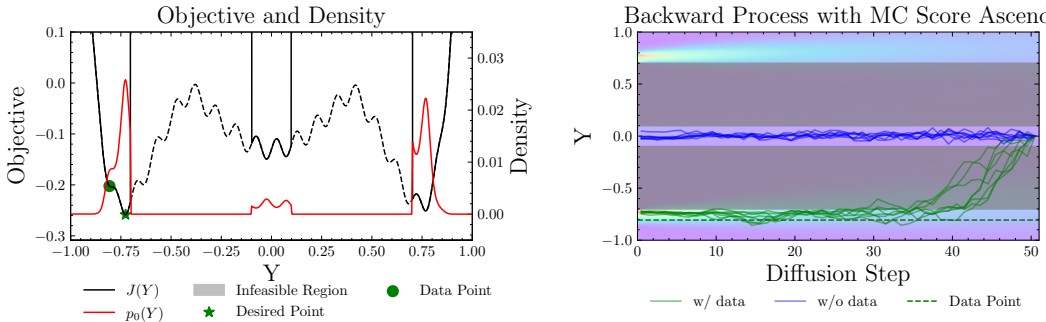

Figure 6: MBD with data vs. without data on a nonconvex function with constraints $||Y| - 0.4| > 0.3$. We want MBD converge to the optimal point ★ with the help of demonstration data ●. Although the demostration point is not optimal, MBD can still converge to the optimal point with the guidance of the demonstration data. Here data serves as a regularization term to guide the diffusion process to the negative optimal point while allowing to use model further to refine the solution.

where demonstrate likelihood term $w_{\text{demo}}(Y^{(0)})$ will draw samples towards data without considering the model. Given $w = w_{\text{demo}}$, $\bar{Y}^{(0)} = \frac{\sum_{Y^{(0)} \in \mathcal{Y}_d^{(i)}} Y^{(0)} p_{\text{demo}}(Y^{(0)})}{\sum_{Y^{(0)} \in \mathcal{Y}_d^{(i)}} p_{\text{demo}}(Y^{(0)})} = Y_{\text{demo}}$. The score function would be

$\nabla_{Y^{(i)}} \log p_i(Y^{(i)}) = -\frac{Y^{(i)}}{1-\bar{\alpha}_i} + \frac{\sqrt{\bar{\alpha}_i}}{1-\bar{\alpha}_i} Y_{\text{demo}}$, which means the score function is a linear combination of the current sample and the demonstration data.

If we don't use Eq. (11) and employ the posterior distribution $p(Y^{(0)}|Y_{\text{demo}}) \propto p_0(Y^{(0)})p_{\text{demo}}(Y^{(0)})$, it will yields update weights $w = w_{\text{demo}}w_{\text{model}}$, which will draw samples to both model and demonstration data. If the demonstration data is not optimal, the final solution will be a compromise between the model and demonstration data. In Fig. 6, the resulted solution will lie between optimal point ★ with the help of demonstration data ●.

Using the max function in $w$ can aviod this issue. In the early stage while $p_J(Y^{(0)})$ is low due to poor sample quality, $w_{\text{demo}}$ will dominate thanks to the high $p_J(Y_{\text{demo}})$. This will draw samples towards the demonstration data as shown in the earlier stage of Fig. 6. As the sample quality improves and $p_J(Y^{(0)}) > p_J(Y_{\text{demo}})$, $w_{\text{model}}$ will dominate and the sample will converge to the optimal point.

### A.5 Experiment Details

#### A.5.1 Simulator and Environment

We leverage the GPU-accelerated simulator Google Brax [20] to design the locomotion and manipulation tasks. All task is set to use positional backend in Brax except for the pushT task, which uses the generalizable backend for better contact dynamics simulation. Here we provide a brief description of each task implementations:

1. **Ant**: The Ant task is a 3D locomotion task where the agent is required to move forward as fast as possible. The reward is composed of the forward velocity of the agent and control cost, same as the original Brax implementation. The control dimension is 8.

2. **Hopper**: The Hopper task is a 2D locomotion task where the agent is required to jumping forward as fast as possible. We use the same reward function as the original Brax implementation. We modify the simulation substeps from 10 to 20 for longer planning horizon given the same control node. The control dimension is 3.

3. **Walker2d**: The Walker2d task is a 2D locomotion task where the agent is required to walk forward. The reward is composed of keep the agent upright and moving forward. The control dimension is 6.

4. **Halfcheetah**: The Halfcheetah task is a 2D locomotion task where the agent is required to run forward. The reward is composed of the forward velocity of the agent and control

cost. We follow the same reward function as the original Brax implementation. The control dimension is 6.

5. **Humanoidrun**: The Humanoidrun task is a 3D locomotion task where the agent is required to run forward. The reward is composed of the forward velocity of the agent and standing upright. Here we also modify the simulation substeps from 10 to 20 for longer planning horizon. The control dimension is 17.

6. **Humanoidstandup**: The Humanoidstandup task is a 3D locomotion task where the agent is required to stand up. The reward is the upright torso position of the agent. The control dimension is 17.

7. **PushT**: The PushT task is a 2D manipulation task where you can apply force to a sphere to push the T-shaped object to the target location. The reward is composed of the distance between the target and the object and orientation difference between the target and the object. To make the task more challenging, we randomize the target location 20cm away from the initial position and make sure the rotational angle is greater than 135 degrees, which makes it hard to solve the task with single continous contact policy. The control dimension is 2.

8. **Car2D**: We implement a 2D car task with standard bicycle dynamics model, where state is $x = [x, y, \theta, v, \delta]$, and action is $u = [a, \delta]$. The dynamics is defined as $\dot{x} = f(x, u) = [v\cos(\theta), v\sin(\theta), \frac{v}{L}\tan(\delta), a, \delta]$. The constraints are defined as the U-shape area in the middle of the map, where the car cannot enter. The reward is composed of the distance between the target and the car and the control cost. The control dimension is 2.

### A.5.2   MBD Hyperparameters

In general, MBD is very little hyperparameters to tune compared with RL. We use the same hyperparameters for all the tasks, with small tweaks for harder tasks.

| Task Name | Horizon | Sample Number | Temperature $\lambda$ |
|---|---|---|---|
| Ant | 50 | 100 | 0.1 |
| Halfcheetah | 50 | 100 | 0.4 |
| Hopper | 50 | 100 | 0.1 |
| Humanoidstandup | 50 | 100 | 0.1 |
| Humanoidrun | 50 | 300 | 0.1 |
| Walker2d | 50 | 100 | 0.1 |
| PushT | 40 | 200 | 0.2 |

Table 4: MBD hyperparameters for various tasks

For diffusion noise schedulling, we use simple linear scheduling $\beta_0 = 1 \times 10^{-4}$ and $\beta_N = 1 \times 10^{-2}$, and the diffusion step number is 100 across all tasks. Each step's $\alpha_i$ is calculated as $\alpha_i = 1 - \beta_i$.

### A.5.3   Baseline Algorithms Implementation

For reinforcement learning implementation, we strictly follow the hyperparameters and implementation details provided by the original Brax repository, which optimize for the best performance. For our self-implemented PushT task, the hyperparameters is ported from Pusher task in Brax for fair comparison. The hyperparameters for the RL tasks are shown in Table 5 and Table 6.

For the zeroth order optimization tasks, we the same hyperparameters as the MBD algorithm.

### A.5.4   Demonstration Collections

For RRT algorithm in Car2D task, we set the max step size to $0.2$, and the max iterations to $1000$ given the maximum episode length is $50$.

For the demonstration collection in Humanoid Jogging task, we first download the mocap data which contains each joints' position in the world frame. Then we use the joint data to calculate the position of torso, thigh and shin position as partial state reference for our task.

| Environment | Algorithm | Timesteps | Reward Scaling | Episode Length |
|---|---|---|---|---|
| Ant | PPO | 100M | 10 | 1000 |
| Hopper | SAC | 6.55M | 30 | 1000 |
| Walker2d | PPO | 50M | 1 | 1000 |
| Halfcheetah | PPO | 50M | 1 | 1000 |
| Pusher | PPO | 50M | 5 | 1000 |
| PushT | PPO | 100M | 1.0 | 100 |
| Humanoidrun | PPO | 100M | 0.1 | 100 |
| Humanoidstandup | PPO | 100M | 0.1 | 1000 |

Table 5: General RL configuration for various environments

| Environment | Minibatches | Updates/Batch | Discounting | Learning Rate |
|---|---|---|---|---|
| Ant | 32 | 4 | 0.97 | $3 \times 10^{-4}$ |
| Hopper | 32 | 4 | 0.997 | $6 \times 10^{-4}$ |
| Walker2d | 32 | 8 | 0.95 | $3 \times 10^{-4}$ |
| Halfcheetah | 32 | 8 | 0.95 | $3 \times 10^{-4}$ |
| Pusher | 16 | 8 | 0.95 | $3 \times 10^{-4}$ |
| PushT | 16 | 8 | 0.99 | $3 \times 10^{-4}$ |
| Humanoidrun | 32 | 8 | 0.97 | $3 \times 10^{-4}$ |
| Humanoidstandup | 32 | 8 | 0.97 | $6 \times 10^{-4}$ |

Table 6: RL specifics for various environments

## A.6   MBD for Online Control

Even though MBD is designed as a trajectory optimization algorithm, it can be naturally extended to receding horizon control as shown in Algorithm 3. By conditioning the planning on each step's observation, MBD's MPC extension can further improve the performance of MBD by $9.6\%$ in terms of reward, especially given control noise as shown in Fig. 7.

The online running frequence of MBD MPC is shown in Table 7 on RTX 4070Ti GPU. Please note that the frequency is calculated under the assumption of solving the whole 50 steps TO problem without reduced model at each iteration. Besides, the Brax code environment we used is not optimized for GPU, so the actual frequency could be higher with optimized environment. In our work we just use Brax as a simple and easy-to-use option. As the major computation time of MBD is spent on the forward dynamics simulation, it can be further improved by using more efficient physics engine.

| Environment | Frequency (Hz) |
|---|---|
| Hopper | 4.56 |
| HalfCheetah | 4.51 |
| Ant | 10.28 |
| Walker2d | 3.49 |
| Humanoidstandup | 6.82 |
| Humanoidrun | 4.03 |

Table 7: Online Running Frequency of receding horizon MBD

## A.7   Sample Number Abalation

Given that sampling-based optimization is the core of MBD, we ablate the sample number in Fig. 8. We can see that MBD converges to the optimal point with as few as 128 samples. The harder the task is, the larger performance gap between MBD and other TO solvers.

**Algorithm 3** Model-based Diffusion with Receding Horizon

1: **Initialize:** Optimize trajectory $x_{0:T}$, $u_{0:T-1}$ with MBD
2: **for** $t = 0$ **to** $t_{\text{final}}$ **do**
3:     Observe the state $x_t$
4:     Optimize trajectory $x_{t:t+T}$, $u_{t:t+T-1}$ with single-step MBD
5:     Apply the first control input $u_t$ to the system
6:     Shift trajectory for next initialization: $x_{t+1:t+T}$, $u_{t+1:t+T-1}$
7: **end for**

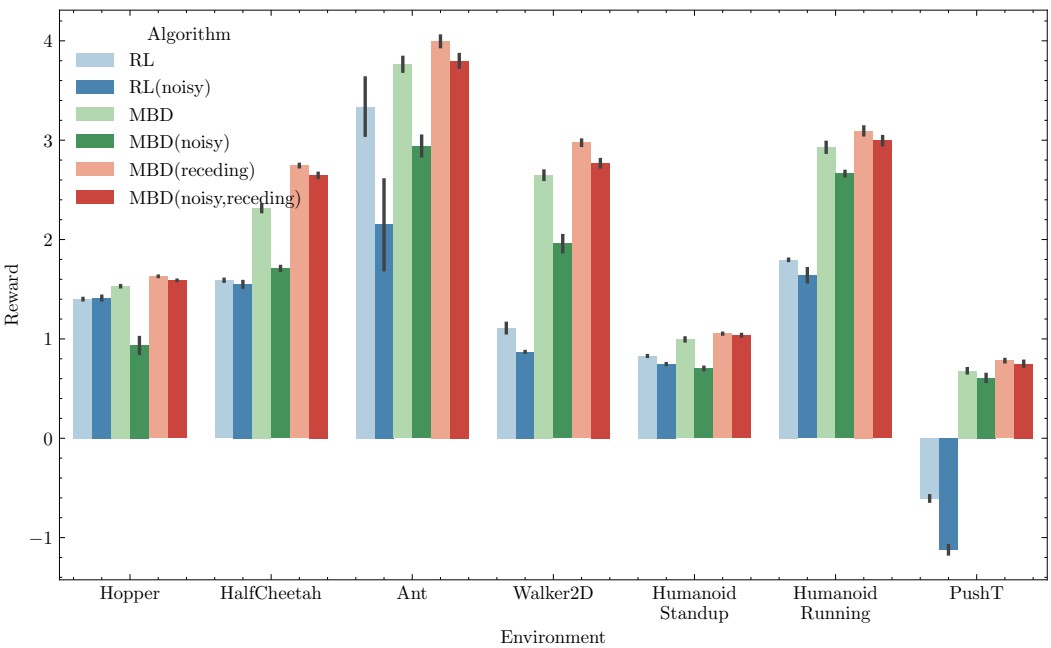

Figure 7: Comparison of the performance between RL, MBD and MBD receding horizon version with both perfect model and noisy model (with $5\%$ control noise). Given perfect model, we find MBD's performance can be further augmented by $9.6\%$ with naive receding horizon planning and leads RL by larger $74.2\%$. Given noisy model, MBD receding horizon version still outperforms RL by $65.3\%$.

### A.8 Objective Function Abalation

Given that RL and TO have different objective settings, especially the horizon difference, we conducted an ablation study by swapping the optimization objectives of MBD and RL. RL optimizes for a longer horizon discounted reward $J = \sum_{t=0}^{H_{\text{RL}}} \gamma^t r_t$, $H = 1000, \gamma < 1$ while MBD optimizes for a shorter horizon undiscounted cumulative rewards $J = \sum_{t=0}^{H_{\text{MBD}}} r_t$, $H = 50$, $\gamma = 1$. Figure 9 shows the performance of MBD and RL under each other's optimization objectives.

MBD outperforms RL by $44.5\%$ under the RL objective and $805.5\%$ under the TO objective. The results demonstrate that MBD's superior performance is attributed to its better diffusion-style iterative optimization process compared with RL's random exploration.

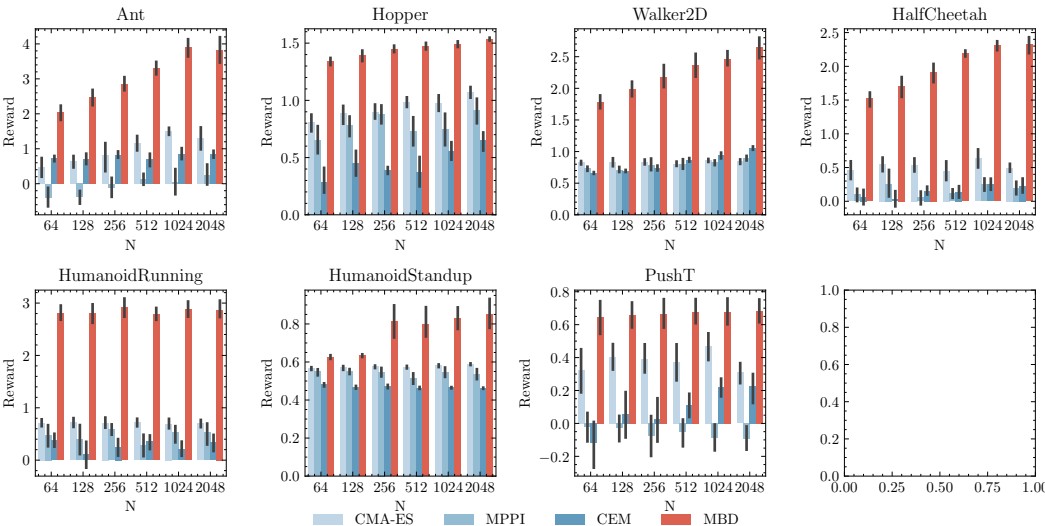

Figure 8: Ablation study on the number of samples's effect on the performance of MBD and other TO solver. For lower-dimensional task like Ant, Hopper, MBD's performance is less sensitive to the number of samples. For higher-dimensional task like Humanoid, MBD's performance is more sensitive to the number of samples but still outperforms all the baselines.


Figure 9: Ablation study on RL and TO objective function. $\gamma = 1, H = 50$ is TO objective, $\gamma < 1, H = 1000$ is RL objective, where $\gamma$ is the discount factor and $H$ is the horizon. For both objective function, MBD outperforms RL by $44.5\%$ and $805.5\%$ on average. We also find RL's overall performance is better given the discounted objective while MBD's performance is better given the total objective, which is consistent with our main results.

  should reflect on how these assumptions might be violated in practice and what the implications would be.

- The authors should reflect on the scope of the claims made, e.g., if the approach was only tested on a few datasets or with a few runs. In general, empirical results often depend on implicit assumptions, which should be articulated.
- The authors should reflect on the factors that influence the performance of the approach. For example, a facial recognition algorithm may perform poorly when image resolution is low or images are taken in low lighting. Or a speech-to-text system might not be used reliably to provide closed captions for online lectures because it fails to handle technical jargon.
- The authors should discuss the computational efficiency of the proposed algorithms and how they scale with dataset size.
- If applicable, the authors should discuss possible limitations of their approach to address problems of privacy and fairness.
- While the authors might fear that complete honesty about limitations might be used by reviewers as grounds for rejection, a worse outcome might be that reviewers discover limitations that aren't acknowledged in the paper. The authors should use their best judgment and recognize that individual actions in favor of transparency play an important role in developing norms that preserve the integrity of the community. Reviewers will be specifically instructed to not penalize honesty concerning limitations.

3. **Theory Assumptions and Proofs**

Question: For each theoretical result, does the paper provide the full set of assumptions and a complete (and correct) proof?

Answer: [Yes]

Justification: This paper includes theoretical results, and all the assumptions and proofs are provided in the main paper.

Guidelines:

- The answer NA means that the paper does not include theoretical results.

- All the theorems, formulas, and proofs in the paper should be numbered and cross-referenced.
- All assumptions should be clearly stated or referenced in the statement of any theorems.
- The proofs can either appear in the main paper or the supplemental material, but if they appear in the supplemental material, the authors are encouraged to provide a short proof sketch to provide intuition.
- Inversely, any informal proof provided in the core of the paper should be complemented by formal proofs provided in appendix or supplemental material.
- Theorems and Lemmas that the proof relies upon should be properly referenced.

4. **Experimental Result Reproducibility**

   Question: Does the paper fully disclose all the information needed to reproduce the main experimental results of the paper to the extent that it affects the main claims and/or conclusions of the paper (regardless of whether the code and data are provided or not)?

   Answer: [Yes]

   Justification: Appendix A.5 provides all the necessary information to reproduce the main experimental results. The code and data are also provided in the supplementary material.

   Guidelines:

   - The answer NA means that the paper does not include experiments.
   - If the paper includes experiments, a No answer to this question will not be perceived well by the reviewers: Making the paper reproducible is important, regardless of whether the code and data are provided or not.
   - If the contribution is a dataset and/or model, the authors should describe the steps taken to make their results reproducible or verifiable.
   - Depending on the contribution, reproducibility can be accomplished in various ways. For example, if the contribution is a novel architecture, describing the architecture fully might suffice, or if the contribution is a specific model and empirical evaluation, it may be necessary to either make it possible for others to replicate the model with the same dataset, or provide access to the model. In general. releasing code and data is often one good way to accomplish this, but reproducibility can also be provided via detailed instructions for how to replicate the results, access to a hosted model (e.g., in the case of a large language model), releasing of a model checkpoint, or other means that are appropriate to the research performed.
   - While NeurIPS does not require releasing code, the conference does require all submissions to provide some reasonable avenue for reproducibility, which may depend on the nature of the contribution. For example
     (a) If the contribution is primarily a new algorithm, the paper should make it clear how to reproduce that algorithm.
     (b) If the contribution is primarily a new model architecture, the paper should describe the architecture clearly and fully.
     (c) If the contribution is a new model (e.g., a large language model), then there should either be a way to access this model for reproducing the results or a way to reproduce the model (e.g., with an open-source dataset or instructions for how to construct the dataset).
     (d) We recognize that reproducibility may be tricky in some cases, in which case authors are welcome to describe the particular way they provide for reproducibility. In the case of closed-source models, it may be that access to the model is limited in some way (e.g., to registered users), but it should be possible for other researchers to have some path to reproducing or verifying the results.

5. **Open access to data and code**

   Question: Does the paper provide open access to the data and code, with sufficient instructions to faithfully reproduce the main experimental results, as described in supplemental material?

   Answer: [Yes]

Justification: The paper provides open access to the data and code, with dedicated README files and instructions to reproduce the main experimental results comes with the supplemental material.

Guidelines:

- The answer NA means that paper does not include experiments requiring code.
- Please see the NeurIPS code and data submission guidelines (`https://nips.cc/public/guides/CodeSubmissionPolicy`) for more details.
- While we encourage the release of code and data, we understand that this might not be possible, so "No" is an acceptable answer. Papers cannot be rejected simply for not including code, unless this is central to the contribution (e.g., for a new open-source benchmark).
- The instructions should contain the exact command and environment needed to run to reproduce the results. See the NeurIPS code and data submission guidelines (`https://nips.cc/public/guides/CodeSubmissionPolicy`) for more details.
- The authors should provide instructions on data access and preparation, including how to access the raw data, preprocessed data, intermediate data, and generated data, etc.
- The authors should provide scripts to reproduce all experimental results for the new proposed method and baselines. If only a subset of experiments are reproducible, they should state which ones are omitted from the script and why.
- At submission time, to preserve anonymity, the authors should release anonymized versions (if applicable).
- Providing as much information as possible in supplemental material (appended to the paper) is recommended, but including URLs to data and code is permitted.

6. **Experimental Setting/Details**

Question: Does the paper specify all the training and test details (e.g., data splits, hyper-parameters, how they were chosen, type of optimizer, etc.) necessary to understand the results?

Answer: [Yes]

Justification: Appendix A.5 provides all the necessary information to understand the experimental results.

Guidelines:

- The answer NA means that the paper does not include experiments.
- The experimental setting should be presented in the core of the paper to a level of detail that is necessary to appreciate the results and make sense of them.
- The full details can be provided either with the code, in appendix, or as supplemental material.

7. **Experiment Statistical Significance**

Question: Does the paper report error bars suitably and correctly defined or other appropriate information about the statistical significance of the experiments?

Answer: [Yes]

Justification: Main quantitative results in Table 2 report the mean and standard deviation over 8 runs given different random seeds.

Guidelines:

- The answer NA means that the paper does not include experiments.
- The authors should answer "Yes" if the results are accompanied by error bars, confidence intervals, or statistical significance tests, at least for the experiments that support the main claims of the paper.
- The factors of variability that the error bars are capturing should be clearly stated (for example, train/test split, initialization, random drawing of some parameter, or overall run with given experimental conditions).
- The method for calculating the error bars should be explained (closed form formula, call to a library function, bootstrap, etc.)

- The assumptions made should be given (e.g., Normally distributed errors).
- It should be clear whether the error bar is the standard deviation or the standard error of the mean.
- It is OK to report 1-sigma error bars, but one should state it. The authors should preferably report a 2-sigma error bar than state that they have a 96% CI, if the hypothesis of Normality of errors is not verified.
- For asymmetric distributions, the authors should be careful not to show in tables or figures symmetric error bars that would yield results that are out of range (e.g. negative error rates).
- If error bars are reported in tables or plots, The authors should explain in the text how they were calculated and reference the corresponding figures or tables in the text.

8. **Experiments Compute Resources**

   Question: For each experiment, does the paper provide sufficient information on the computer resources (type of compute workers, memory, time of execution) needed to reproduce the experiments?

   Answer: [Yes]

   Justification: We provide the hardware requirements in Appendix A.5.1.

   Guidelines:
   - The answer NA means that the paper does not include experiments.
   - The paper should indicate the type of compute workers CPU or GPU, internal cluster, or cloud provider, including relevant memory and storage.
   - The paper should provide the amount of compute required for each of the individual experimental runs as well as estimate the total compute.
   - The paper should disclose whether the full research project required more compute than the experiments reported in the paper (e.g., preliminary or failed experiments that didn't make it into the paper).

9. **Code Of Ethics**

   Question: Does the research conducted in the paper conform, in every respect, with the NeurIPS Code of Ethics https://neurips.cc/public/EthicsGuidelines?

   Answer: [Yes]

   Justification: The research conducted in the paper conforms with the NeurIPS Code of Ethics.

   Guidelines:
   - The answer NA means that the authors have not reviewed the NeurIPS Code of Ethics.
   - If the authors answer No, they should explain the special circumstances that require a deviation from the Code of Ethics.
   - The authors should make sure to preserve anonymity (e.g., if there is a special consideration due to laws or regulations in their jurisdiction).

10. **Broader Impacts**

    Question: Does the paper discuss both potential positive societal impacts and negative societal impacts of the work performed?

    Answer: [NA]

    Justification: The paper does not address societal impact.

    Guidelines:
    - The answer NA means that there is no societal impact of the work performed.
    - If the authors answer NA or No, they should explain why their work has no societal impact or why the paper does not address societal impact.
    - Examples of negative societal impacts include potential malicious or unintended uses (e.g., disinformation, generating fake profiles, surveillance), fairness considerations (e.g., deployment of technologies that could make decisions that unfairly impact specific groups), privacy considerations, and security considerations.

- The conference expects that many papers will be foundational research and not tied to particular applications, let alone deployments. However, if there is a direct path to any negative applications, the authors should point it out. For example, it is legitimate to point out that an improvement in the quality of generative models could be used to generate deepfakes for disinformation. On the other hand, it is not needed to point out that a generic algorithm for optimizing neural networks could enable people to train models that generate Deepfakes faster.
- The authors should consider possible harms that could arise when the technology is being used as intended and functioning correctly, harms that could arise when the technology is being used as intended but gives incorrect results, and harms following from (intentional or unintentional) misuse of the technology.
- If there are negative societal impacts, the authors could also discuss possible mitigation strategies (e.g., gated release of models, providing defenses in addition to attacks, mechanisms for monitoring misuse, mechanisms to monitor how a system learns from feedback over time, improving the efficiency and accessibility of ML).

11. **Safeguards**

Question: Does the paper describe safeguards that have been put in place for responsible release of data or models that have a high risk for misuse (e.g., pretrained language models, image generators, or scraped datasets)?

Answer: [NA]

Justification: The paper does not release data or models that have a high risk for misuse.

Guidelines:

- The answer NA means that the paper poses no such risks.
- Released models that have a high risk for misuse or dual-use should be released with necessary safeguards to allow for controlled use of the model, for example by requiring that users adhere to usage guidelines or restrictions to access the model or implementing safety filters.
- Datasets that have been scraped from the Internet could pose safety risks. The authors should describe how they avoided releasing unsafe images.
- We recognize that providing effective safeguards is challenging, and many papers do not require this, but we encourage authors to take this into account and make a best faith effort.

12. **Licenses for existing assets**

Question: Are the creators or original owners of assets (e.g., code, data, models), used in the paper, properly credited and are the license and terms of use explicitly mentioned and properly respected?

Answer: [Yes]

Justification: We mentioned the package used in the paper on Appendix A.5.

Guidelines:

- The answer NA means that the paper does not use existing assets.
- The authors should cite the original paper that produced the code package or dataset.
- The authors should state which version of the asset is used and, if possible, include a URL.
- The name of the license (e.g., CC-BY 4.0) should be included for each asset.
- For scraped data from a particular source (e.g., website), the copyright and terms of service of that source should be provided.
- If assets are released, the license, copyright information, and terms of use in the package should be provided. For popular datasets, `paperswithcode.com/datasets` has curated licenses for some datasets. Their licensing guide can help determine the license of a dataset.
- For existing datasets that are re-packaged, both the original license and the license of the derived asset (if it has changed) should be provided.

- If this information is not available online, the authors are encouraged to reach out to the asset's creators.

13. **New Assets**

Question: Are new assets introduced in the paper well documented and is the documentation provided alongside the assets?

Answer: [NA]

Justification: The paper does not introduce new assets.

Guidelines:

- The answer NA means that the paper does not release new assets.
- Researchers should communicate the details of the dataset/code/model as part of their submissions via structured templates. This includes details about training, license, limitations, etc.
- The paper should discuss whether and how consent was obtained from people whose asset is used.
- At submission time, remember to anonymize your assets (if applicable). You can either create an anonymized URL or include an anonymized zip file.

14. **Crowdsourcing and Research with Human Subjects**

Question: For crowdsourcing experiments and research with human subjects, does the paper include the full text of instructions given to participants and screenshots, if applicable, as well as details about compensation (if any)?

Answer: [NA]

Justification: The paper does not involve crowdsourcing nor research with human subjects.

Guidelines:

- The answer NA means that the paper does not involve crowdsourcing nor research with human subjects.
- Including this information in the supplemental material is fine, but if the main contribution of the paper involves human subjects, then as much detail as possible should be included in the main paper.
- According to the NeurIPS Code of Ethics, workers involved in data collection, curation, or other labor should be paid at least the minimum wage in the country of the data collector.

15. **Institutional Review Board (IRB) Approvals or Equivalent for Research with Human Subjects**

Question: Does the paper describe potential risks incurred by study participants, whether such risks were disclosed to the subjects, and whether Institutional Review Board (IRB) approvals (or an equivalent approval/review based on the requirements of your country or institution) were obtained?

Answer: [NA]

Justification: The paper does not involve crowdsourcing nor research with human subjects.

Guidelines:

- The answer NA means that the paper does not involve crowdsourcing nor research with human subjects.
- Depending on the country in which research is conducted, IRB approval (or equivalent) may be required for any human subjects research. If you obtained IRB approval, you should clearly state this in the paper.
- We recognize that the procedures for this may vary significantly between institutions and locations, and we expect authors to adhere to the NeurIPS Code of Ethics and the guidelines for their institution.
- For initial submissions, do not include any information that would break anonymity (if applicable), such as the institution conducting the review.

