# OpenReview forum: "Model-based Diffusion for Trajectory Optimization"
_NeurIPS.cc/2024/Conference — NeurIPS 2024 poster_

### Official Review · Reviewer_p88D · 2024-07-06

**Soundness:** 3
**Presentation:** 3
**Contribution:** 3
**Rating:** 5
**Confidence:** 2

**Summary:**

The paper proposed MBO, a novel method to solve trajectory optimization problems with diffusion models. MBD formulates the optimization problem as a sampling problem and estimates the score function using the dynamic model.

**Strengths:**

- The motivation of the paper is valid. In the trajectory optimization problem, we already know the underlying dynamics of the environment. Therefore, it is natural to utilize such information when we use diffusion models to solve trajectory optimization problems.

- The problem statement is clearly written and easy to follow, even for readers who are not familiar with the topic.

- Experiment results demonstrate that MBD outperforms prior methods in various domains.

**Weaknesses:**

- In the introduction, the authors claimed that existing diffusion-based approaches often require high-quality demonstration data. However, I cannot find any evidence of experiment results supporting this claim in the manuscript. Diffusion-based planners are not in the baselines of the experiments, and there are several model-free diffusion-based methods that can efficiently generate dynamically feasible and high-rewarding trajectories in RL fields. Could authors validate the reason why model-free diffusion planners are not included in the baselines?

- The authors also claimed that one of the advantages of model-based diffusion over model-free diffusion is its generalization ability (e.g., generalization to new tasks). However, I cannot find empirical evidence for this claim. Could the authors elaborate more on this claim?

**Questions:**

There are some minor questions for better understanding.

- For model-free diffusion, we need to train the score function estimator through the data. In model-based diffusion, which part should be parametrized by the neural network and trained with the data?

- MBD is compared with the baselines, including zeroth-order optimization and RL algorithms. Is there any other recently proposed method for trajectory optimization?

- For data-augmented MBD, there are also several works that utilize model-free diffusion models to augment data given sub-optimal offline datasets. They claim that through stitching ability, model-free diffusion can also generate novel trajectories [1, 2]. Could authors validate empirically that MBD is more powerful compared to model-free diffusion models given partial and dynamically infeasible datasets?

[1] Ajay, Anurag, et al. "Is Conditional Generative Modeling all you need for Decision Making?." The Eleventh International Conference on Learning Representations.

[2] Li, Guanghe, et al. "DiffStitch: Boosting Offline Reinforcement Learning with Diffusion-based Trajectory Stitching." Forty-first International Conference on Machine Learning.

**Limitations:**

- Please see the weaknesses and questions above.

---

> ### Author Rebuttal · Authors · 2024-08-07
>
> We thank reviewer for the comments and suggestions. We address the comments and suggestions as follows:
>
> ---
>
> > In the introduction, the authors claimed that existing diffusion-based approaches often require high-quality demonstration data. However, I cannot find any evidence of experiment results supporting this claim in the manuscript.
>
> Sorry for the confusion. As we shown in Sec 5.1, all the tasks is solved by MBD from scratch without any demonstration data, which is the key distinction between MBD and MFD.
>
> ---
>
> > Could authors validate the reason why model-free diffusion planners are not included in the baselines?
>
> We are not comparing with MFD because **MFD is not a direct TO solver but a model-free imitation learning method**.
> For input, MFD needs large amount of high-quality demonstration data to learn a trajectory.
> In contrast, MBD is a model-based TO solver that can address TO problems from scratch without relying on demonstration data. Even in cases where demonstrations are used, as discussed in Section 4.3, these demonstrations are partial, feasible, and limited in quantity. Essentially, it is impractical to train a policy effectively with such sparse and low-quality data.
> In short, the problem setting difference makes comparison between MFD and MBD not meaningful.
>
> Please refer to General Response 3 for more details.
>
> ---
>
> > The authors also claimed that one of the advantages of model-based diffusion over model-free diffusion is its generalization ability (e.g., generalization to new tasks). However, I cannot find empirical evidence for this claim. Could the authors elaborate more on this claim?
>
> The generalization ability of MBD comes from the fact that MBD *solve* the trajectory given any model and cost function, while MFD *imitate* the trajectory given the demonstration data.
> MBD takes the model and cost function as input, and generate the trajectory satisfying the model and optimal cost function.
> In contrast, MFD takes the demonstration data as input, and generate the trajectory from the demonstration distribution.
>
> In other words, for the same modification to the task or dynamics, MBD can re-solve the trajectory with the same algorithm, while MFD needs to re-train the network with the newly collected data under the new setting.
>
> ---
>
> > For model-free diffusion, we need to train the score function estimator through the data. In model-based diffusion, which part should be parametrized by the neural network and trained with the data?
>
> Please refer to general response 3 for the training process clarification. Generally speaking, MBD could be training-free but can be further augmented with data and training.
>
> ---
>
> > MBD is compared with the baselines, including zeroth-order optimization and RL algorithms. Is there any other recently proposed method for trajectory optimization?
>
> Solving TO given discontinous hybrid dynamics is generally considered hard, especially for traditional nonlinear programming based solvers like interior point methods, barrier methods, and gradient-based methods which requires the gradient/Hessian information of the dyanmics and costs.
> To the best of our knowledge, **MBD is the first TO method that can solve full TO problems for high-dimensional tasks with hybrid dynamics** thanks to the diffusion-style optimization process.
> Those tasks are used to be considered to be solved by RL algorithms with a large amount of data.
> That's why we not only compare with other TO methods, but also add RL as a performance reference.
>
> ---
>
> > For data-augmented MBD, there are also several works that utilize model-free diffusion models to augment data given sub-optimal offline datasets. They claim that through stitching ability, model-free diffusion can also generate novel trajectories [1, 2]. Could authors validate empirically that MBD is more powerful compared to model-free diffusion models given partial and dynamically infeasible datasets?
>
> **Comparing MBD with MFD is not apples-to-apples comparison due to their setting difference**.
> The major difference lies in the problem setting: MBD is a TO solver while [1,2] is offline RL algorithms.
> In other words, in terms of data availability, MBD does not require any data (but can be augmented with data), while [1,2] require large amount of offline dataset like D4RL.
> Besides, the dataset like D4RL used in [1,2] is always dynamically feasible (even though it might not be optimal), while MBD can easily handle dynamically infeasible trajectory.
> In our experiments, the demo data is not dynamically feasible so [1,2] can not even be applied, whereas MBD successfully integrates the demo data.
> In short, it is not meaningful to compare MBD with [1,2] due to the difference in input and the data requirement of MBD is much weaker than [1,2].

---

> > ### Comment · Reviewer_p88D · 2024-08-13
> >
> > Thank you for the authors' detailed response. I understand that the settings of MFD and MBD are quite different, but it might be better to empirically demonstrate that MBD can generalize to various dynamics and cost functions. Therefore, I maintain the score.

---

> ### Author Response · Authors · 2024-08-13
> **Thanks for the feedback!**
>
> We appreciate the reviewer's suggestion to empirically demonstrate the generalizability of MBD to various dynamics and cost functions. As shown in Sec.5 of the paper, we have conducted extensive experiments, which suggest that MBD, as a **Trajectory Optimization solver**  (aka planning, a very important problem in decision making and RL), inherently adapts to **ANY** given model and cost function. Alongside with its model-based nature as a Trajectory Optimization, we hope it's sufficient to showcast its capacity to handle diverse dynamics and costs.
>
> However, we would like to seek further clarification on reviewer's specific concerns regarding the generalization abilities of MBD. If there are particular aspects or additional experiments you believe would more effectively demonstrate this capability, we would be delighted to explore these avenues to address your concerns.

---

### Official Review · Reviewer_YGTC · 2024-07-12

**Soundness:** 3
**Presentation:** 3
**Contribution:** 3
**Rating:** 5
**Confidence:** 3

**Summary:**

This work introduces model-based diffusion (MBD) to solve trajectory optimization (TO) problems without relying on data. The key idea is to explicitly compute the score function by leveraging the model information in TO problems. This paper also shows that MBD has interesting connections to sampling-based optimization. The empirical evaluations demonstrate that MBD outperforms state-of-the-art TO methods.

**Strengths:**

- The model-based diffusion framework enables an effective trajectory planner.
- This algorithm shows good performance on various tasks compared to other state-of-the-art algorithms.

**Weaknesses:**

- The theoretical derivation and notation in the methodology are quite confusing and hard to follow.
- The comparison to some simple baseline of RL, such as PPO, cannot demonstrate the real advantages.
- In line 137 and equation $(3)$, there may be an incorrect formula:
$$
p_{i|i-1}(\cdot, Y^{(i-1)}) \sim \mathcal{N}(\sqrt{\alpha_i}Y^{(i-1)},1-\alpha_iI)
$$
and
$$
p_{i|0}(\cdot, Y^{(0)}) \sim \mathcal{N}(\sqrt{\overline{\alpha}_i}Y^{(0)},1-\overline{\alpha}_iI)
$$

- In equation $(8)$, there may be an incorrect formula:
$$
\phi_i(Y^{(0)})\propto\cdots\propto\mathcal{N}(Y^{(0)}-\frac{Y^{(i)}}{\sqrt{\overline{\alpha}_i}},\frac{I}{\overline{\alpha}_i}-I)
$$

- In all algorithms, it is evident that the calculation step can be simplified:

$Y^{(i-1)} \leftarrow \frac{1}{\sqrt{\alpha_i}}(Y^{(i)}-Y^{(i)}+\sqrt{\overline{\alpha}_i}\overline{Y}^{(0)}(\mathcal{Y}^{(i)})) = \frac{\sqrt{\overline{\alpha}_i}}{\sqrt{\alpha_i}}\overline{Y}^{(0)}(\mathcal{Y}^{(i)})=\sqrt{\overline{\alpha}_{i-1}}\overline{Y}^{(0)}(\mathcal{Y}^{(i)})$

Note that this coefficient $\sqrt{\overline{\alpha}_{i-1}}$ is canceled at the first step. Also, note that in equations $(9b)$, $(10d)$, and $(13)$, it is a weighted average of the samples from $\mathcal{Y}^{(i)}$. It may be clearer if the calculation steps are more simplified and easy to understand.

- In Table 3, the computational times of the CEM and MC-MBD algorithms are close. I'm wondering if this table includes the computation time for environment interaction. It might be fairer to compare just the computation times for the algorithms.

**Questions:**

Please refer to the Weaknesses.

**Limitations:**

Limitations are not discussed in the manuscript.

---

> ### Author Rebuttal · Authors · 2024-08-07
>
> We thank reviewer for the comments and suggestions. We address the comments and suggestions as follows:
>
> ---
>
> > The theoretical derivation and notation in the methodology are quite confusing and hard to follow.
>
> Our approach utilizes Reverse SDE, a technique commonly used in diffusion models, to optimize trajectories under an annealed schedule. Similar to Model Free Diffusion, estimating the score function is crucial. Instead of learning from large data sets, we use a novel sampling strategy. We employ importance sampling with a Gaussian proposal distribution centered on the current $Y^{(i)}$, which improves the use of model information and enhances our model's efficiency and accuracy in dynamic settings.
>
> Here we provide some more details on the purpose of each step in the derivation. Eq. (7) is applying the gradient rule to the diffused distribution; from (7) to (9), the goal is to try to rewrite the integral in (7) as an expectation on $Y^{(0)}$, for which purpose we need to rewrite the density function $p_{i|0}$ as a samplable proposal density function for $Y^{(0)}$ (which is (8)). Lastly, in eq. (9) we replace the expectation with a Monte-Carlo sampling of $Y^{(0)}$. We also provicde a more detailed Notation Table below. We hope these helps and we would appreciate if you can provide more specific suggestions on improving the presentation.
>
>
> Table5: Notation Table
> | Meaning | Symbol |
> |--|--|
> | state, control at time t  | $x_t,u_t$|
> | state control pair at time t | $y_t$ |
> | diffused random variable at step i | $Y^{(i)}$ |
> | density of diffused distribution at step i | $p_i(\cdot)$ |
> | density of diffused r.v. at step i conditioned on step j| $p_{i\mid j}(\cdot \mid \cdot)$ |
> | samples collected from proposal distribution at setp i | $\mathcal{Y}^{(i)}$ |
> | scale down factor at step i | $\alpha_i$
> | accumulated scale down factor at step i | $\bar{\alpha}_i$ |
> | dynamic feasibility density, optimality density, constraint density | $p_d(\cdot), p_J(\cdot), p_g(\cdot)$ |
>
>
> ---
>
> > The comparison to some simple baseline of RL, such as PPO, cannot demonstrate the real advantages.
>
> For the reason why use model-free RL as one performance reference, please refer to general response 1.
> Generally speaking, due to the complexity of the task and non-smoothness of the dynamics, classic gradient-based optimization algorithms like CEM and CMA-ES failed to solve the task.
> For the reference purpose, we include model-free RL algorithms like PPO and SAC, which are generally considered to be state-of-the-art in such tasks [2,3], into the comparison.
> Our implementation of PPO and SAC are adopted on the Google Deepmind Brax [1], which involves common improvements like large-scale parallelism, reward scaling, hyperparameter sweeps and state normalization, etc.
> So we believe that the our well-tuned implementation of PPO and SAC can provide a strong reference to demonstrate the effectiveness of our proposed method.
>
> [1] C. D. Freeman, E. Frey, A. Raichuk, S. Girgin, I. Mordatch, and O. Bachem, “Brax -- A Differentiable Physics Engine for Large Scale Rigid Body Simulation,” Jun. 24, 2021, arXiv: arXiv:2106.13281.
>
> [2] S. Çalışır and M. K. Pehlivanoğlu, “Model-Free Reinforcement Learning Algorithms: A Survey,” in 2019 27th Signal Processing and Communications Applications Conference (SIU), Apr. 2019, pp. 1–4. doi: 10.1109/SIU.2019.8806389.
>
> [3] F. E. Dorner, “Measuring Progress in Deep Reinforcement Learning Sample Efficiency,” Feb. 09, 2021, arXiv: arXiv:2102.04881.
>
> ---
>
> > In line 137 and equation $(3)$, there may be an incorrect formula. In equation $(8)$, there may be an incorrect formula
>
> Yes, we made a typo in the equation $(3)$ and $(8)$ by replacing variance with standard deviation. We will correct the typos in the revised manuscript. Please note that this does not affect other derivations in the paper and all the subsequent equations are correct. Thank you for pointing out the errors.
>
> ---
>
> > Note that this coefficient $\sqrt{\overline{\alpha}_{i-1}}$ is canceled at the first step. Also, note that in equations $(9b)$, $(10d)$, and $(13)$, it is a weighted average of the samples from $\mathcal{Y}^{(i)}$. It may be clearer if the calculation steps are more simplified and easy to understand.
>
> That's a quite insightful observation.
> We would like to point out that **there are two perspectives to interpret the calculation steps**.
> For the first perspective, we first calculate the score function and then plug it into the reverse process, which is the standard way to derive the backward process in the diffusion model.
> For the second perspective, as we mentioned in Sec 4.1 *Connection with Sampling-based Optimization*, plugging in the score actually leads to an interesting connection with existing optimization algorithms in single-step diffusion case, which coincides with the reveiwer's opinion here.
>
> We keep the score expression in the main text for the sake of completeness, but we will provide a alternative version in the appendix.
> Thank you for the suggestion!
>
> ---
>
> > In Table 3, the computational times of the CEM and MC-MBD algorithms are close. I'm wondering if this table includes the computation time for environment interaction. It might be fairer to compare just the computation times for the algorithms.
>
> Yes, the computational times in Table 3 include the computation time for environment interaction. The reason why their computational time is close is that we match the iteration number of baseline algorithms to MBD diffusion steps for fair comparison, which is a common practice for the planning algorithm used in MBRL. We will clarify this in the appendix.

---

> ### Comment · Reviewer_YGTC · 2024-08-13
>
> Thanks for your response. However, I still have some concerns about the comparison baselines. PPO and SAC can provide some reference for the performance, but they are not strong baselines. Recent model-free RL methods such as DIPO [1], QSM [2] show much better performance compared to PPO and SAC. Besides, since this method is model-based Diffusion, why not compare it to  model-based RL baselines, such as Dream-v3 [3], TD-MPC2 [4].
>
> [1] Long Yang, et al, Policy Representation via Diffusion Probability Model for Reinforcement Learning, arXiv:2305.13122.
>
> [2] Michael Psenka, et al, Learning a Diffusion Model Policy from Rewards via Q-Score Matching, ICML 2024.
>
> [3] D Hafner, et al, Mastering Diverse Domains through World Models, arXiv:2301.04104.
>
> [4] Nicklas Hansen, et al, TD-MPC2: Scalable, Robust World Models for Continuous Control, ICLR 2024.

---

> ### Author Response · Authors · 2024-08-13
> **Thanks for the feedback!**
>
> Thanks for the suggestion.
>
> Fistly, comparing with MBRL, our setting centers on Trajectory Optimization, which assume model is available. In contrast, model-based RL baselines like Dream-v3 and TD-MPC2 are designed to concurrently learn the model and the policy. Therefore, a direct comparison is not entirely appropriate as it isn’t an apples-to-apples situation. However, examining the optimizer of TD-MPC2[4], one can observe that the planner module used for solving optimization tasks with learned dynamics is MPPI, which we also employ as a baseline across all our environments. In fact, MBD could be regarded as a subroutine within model-based RL algorithms, capable of optimizing the trajectory given a model. We believe this integration offers a promising avenue for future research.
>
> Secondly, comparing with more advanced MFRL, we think PPO and SAC are still representative baselines for the asymptotic performance.
> Specifically, DIPO and QSM demonstrate better sample efficiency and multi-modal representation compared with PPO and SAC, but since we train RL util converge, we only focus on asymptotic performance, both of which does not show significant improvement over PPO/SAC.
> If we are proposing a diffusion-based RL algorithm, we would definitely compare with the diffusion-based RL algorithm like DIPO and QSM.
> However, since MBD is a trajectory optimization algorithm, we believe our choice of PPO and SAC as the aymptotic performance reference would be enough to demonstrate the effectiveness of our method, especially given the fact that there are no existing TO algorithm can solve those task well.
>
> If you have any further suggestions or concerns, please let us know. We appreciate your feedback!

---

> > ### Comment · Reviewer_YGTC · 2024-08-14
> >
> > Thanks for the clarification. I will raise my score.

---

### Official Review · Reviewer_X3LL · 2024-07-13

**Soundness:** 3
**Presentation:** 3
**Contribution:** 2
**Rating:** 5
**Confidence:** 4

**Summary:**

This paper presents Model Based Diffusion (MBD), a novel approach to trajectory optimization that leverages a diffusion process. Unlike traditional diffusion models where denoising networks are trained on data using a score matching loss, MBD directly computes the score function for a given target distribution. Specifically, for a trajectory optimization problem, the unnormalized target distribution is formulated as a product of three terms: an optimality term that depends on the cost function, a dynamical feasibility term, and a constraint satisfaction term. The authors demonstrate that in each iteration of the reverse diffusion process, the score function can be approximated using Monte-Carlo estimation – a weighted sum of samples drawn from a Gaussian distribution parameterized by the current trajectory, where the weight of each sample is the target density evaluated at that sample. Additionally, they show how their approach can be extended to handle scenarios with noisy observations of trajectories from the target distribution. Finally, they present experimental results on locomotion and manipulation tasks, demonstrating that their method outperforms other trajectory refinement techniques and RL algorithms in both average step reward and computational time.

**Strengths:**

This paper introduces an innovative approach that leverages diffusion models directly as solvers for trajectory optimization. The key advantage is that if the cost function, dynamics, and constraints can be specified for a given problem, the proposed iterative refinement approach can obtain samples from the desired distribution without having to rely on any demonstration data. If demonstration data is available, the authors demonstrate how it could be used to augment their approach.
The paper is also well-presented and easy to understand.

**Weaknesses:**

The requirement for explicit specification of the cost function, dynamics, and constraints can be a limitation in real-world applicatioms. It would also be interesting to see how MBD would perform in the sparse reward or goal completion settings.
The performance of MBD presumably heavily relies on the accuracy of the provided dynamics model. Inaccurate models could lead to suboptimal trajectories and compounding errors. It would be useful to study how robust MBD is to varying degrees of model inaccuracy.

**Questions:**

- The paper primarily evaluates MBD on tasks with relatively short horizons (e.g., 50 steps for most locomotion tasks). It would be valuable to understand how the performance of MBD scales with increasing horizon lengths. Do the computational benefits over RL persist for longer horizons?
- How does the performance of MBD vary as a function of the number of samples used in the Monte Carlo approximation?
- For the RL agents, it looks like the full episode length (e.g. 1000 steps for locomotion tasks like hopper, halfcheetah) is used. The RL agents are presumably trained to maximize rewards over the entire episode, while is MBD optimized for a much shorter horizon. Is this then a fair comparison? The paper could benefit from a discussion of this comparison or additional experiments with RL agents trained on shorter horizons.
- How do you envision incorporating conditioning signals in MBD? This would be important for adapting MBD to a receding horizon strategy, where the trajectory is re-planned at each time step based on new observations.

**Limitations:**

The authors have not discussed the limitations of their work.

---

> ### Author Rebuttal · Authors · 2024-08-07
>
> We thank reviewer for the comments and suggestions. We address the comments and suggestions as follows:
>
> ---
>
> > The requirement for explicit specification of the cost function, dynamics, and constraints can be a limitation in real-world applications.
>
> The requirement of dynamics and cost is defined by the nature of the TO problem we studied in the paper, i.e. given a model and cost function, how to solve the optimal control problem efficiently.
> The usage of dynamic model and cost function actually could be one key advantage of MBD compared with MFD.
> For in most robotics tasks including manipulator control, locomotion, and navigation, we have access to the first principle model of the system and the cost function is usually designed by the user.
> Our method makes full use of the model and cost function to enable a general and efficient trajectory optimization algorithm without training from demonstration data for each platform.
>
> But we totally agree with the reviewer that in some cases, accurate model and objective function could be hard to obtain, which can be further addressed by model learning methods as we mentioned in general response 3.
>
> ---
>
> > It would also be interesting to see how MBD would perform in the sparse reward or goal completion settings.
>
> Actually, our `pushT` and `Car UMaze Navigation` tasks are exactly the sparse reward or goal completion settings.
> For `Car UMaze Navigation` task, the agent only gets reward when it reaches the goal, which make standard sampling-based algorithm like MPPI fails due to the exploration difficulty.
> Same for the `pushT` task, the agent only gets reward when pushing the object overlapping with the random goal position and even large scale RL algorithm like PPO cannot solve it due to the sparsity reward and goal completion settings.
> `pushT` task are used to be solved by complex convex decomposition method [1] or imitation learning method [2].
> MBD is the first sampling-based algorithm that can solve these tasks efficiently.
>
> **The superior performance of MBD in sparse-reward tasks comes from the forward noise injection process, which make the sparse reward spread to larger space and make the goal completion more reachable**.
> In the early diffusion step, larger noise is convoluted into the objective function, which helps to makes it easier for MBD to find the space where the goal is reachable.
> In the later smaller noise stage, MBD can further refine the trajectory to reach the goal.
>
> [1] B. P. Graesdal et al., “Towards Tight Convex Relaxations for Contact-Rich Manipulation,” Jul. 05, 2024, arXiv: arXiv:2402.10312
>
> [2] C. Chi et al., “Diffusion Policy: Visuomotor Policy Learning via Action Diffusion,” Jun. 01, 2023, arXiv: arXiv:2303.04137
>
> ---
>
> > Inaccurate models could lead to suboptimal trajectories and compounding errors. It would be useful to study how robust MBD is to varying degrees of model inaccuracy.
>
> Please refer to general response 4 for the robustness of MBD to noisy dynamics.
> We also tested MBD's performance under weight-sensitive task like humanoidstandup. We plan/train the trajectory with nominal model while test it with a model with 20% mass perturbation as shown in Table 4.
> This again highlights the robustness of MBD thanks to the iterative refinement process in the optimization, which make the solution less sensitive to the model inaccuracy.
>
> Table 4: Mean Reward of HumanoidStandup
> | Method | RL | MBD | MBD (Receding Horizon) |
> |-|-|-|-|
> | Nominal Model | 0.83 | 0.99 | 1.05 |
> | 20% Mass Perturbation | 0.72 | 0.84 | 0.89 |
>
>
> Lastly, we want to emphasize that MBD is a TO solver that leverages the model information, which means MBD can adapt quickly to the model inaccuracy by updating the trajectory with the new model.
> This is a key advantage of MBD compared with MFD, which requires recollecting data and retraining the model given new dynamics.
>
> ---
>
> > The paper primarily evaluates MBD on tasks with relatively short horizons. It would be valuable to understand how the performance of MBD scales with increasing horizon lengths. Do the computational benefits over RL persist for longer horizons?
>
> Please refer to general response 1 for optimizing the RL objective with MBD. MBD still outperform RL by $44.5\%$ when horizon length is increased to 1000 steps.
>
> ---
>
> > How does the performance of MBD vary as a function of the number of samples used in the Monte Carlo approximation?
>
> Thanks for the suggestion and we will include the ablation study of the number of samples in the Monte Carlo approximation in the revised version. Reviewer can refer to Figure 3 in the accompanying PDF.
> The results indicated that MBD is less sensitive to the number of samples compared with other baselines and can handle most tasks well even with 128 samples.
>
> ---
>
> > The RL agents are presumably trained to maximize rewards over the entire episode, while is MBD optimized for a much shorter horizon. Is this then a fair comparison?
>
> We thank the reviewer for the insightful question. Please refer to general response 1 for the detailed explanation of comparing MBD with RL by swaping the optimization objective.
> In short, the comparision is fair due to RL objective has a discount factor. MBD can outperform RL in both short and long horizon settings.
>
> ---
>
> > How do you envision incorporating conditioning signals in MBD? This would be important for adapting MBD to a receding horizon strategy, where the trajectory is re-planned at each time step based on new observations.
>
> MBD can be easily extended into the online setting by adopting a simple receding horizon strategy in general MPC algorithms.
> Please refer to general response 2 for further details about the extension of MBD to online setting.
> In short, by combing with the most naive receding horizon strategy, MBD can finetune the original plan at each step easily without modification to the core algorithm.

---

> > ### Comment · Reviewer_X3LL · 2024-08-12
> > **Thank you for your response**
> >
> > Thank you for your response to my questions and for conducting the additional experiments.
> >
> > I have a few follow-up questions:
> >
> > Regarding the extension to closed-loop control with a receding horizon, could you elaborate on how the current state x_t
> > is incorporated into Algorithm 1?
> >
> > In lines 506-507, you mention that for pushT, the reward is composed of the distance between the target and the object, as well as the orientation difference between them. Is the reward signal available at each state and computed based on the distance and orientation, or is it a sparse reward provided only when the distance/orientation are below certain thresholds?

---

> > > ### Author Response · Authors · 2024-08-13
> > > **Thanks for the comments!**
> > >
> > > Thank you for the comments. Here are our responses:
> > >
> > > > Regarding the extension to closed-loop control with a receding horizon, could you elaborate on how the current state x_t is incorporated into Algorithm 1?
> > >
> > > We would like to refer to line 4 on Algorithm 2, where Y represents the entire control input sequence that $ Y = U = u_{1:T} $. Given the current state $x_t$, we can roll out the control sequences starting from $x_t$ and generate the predicted state sequence $x_{t+1:t+T}$. This procedure allows us to incorporate and current state $x_t$ into MBD's framework just by calculating the score function as shown in line 5 on Algorithm 2. The closed-loop process is effectively streamlined, as our non-receding-horizon MBD algorithm is capable of initiating from any initial state without requiring conditional training, as demonstrated in Algorithm 2, thanks to the model-based nature of MBD. We hope this would clarify the reviewer's inquiry.
> > >
> > > ---
> > >
> > > > In lines 506-507, you mention that for pushT, the reward is composed of the distance between the target and the object, as well as the orientation difference between them. Is the reward signal available at each state and computed based on the distance and orientation, or is it a sparse reward provided only when the distance/orientation are below certain thresholds?
> > >
> > > Thanks for the clarification question. The reward signal is computed at each state without certain thereshold to the object. We consider this reward signal as a sparse since reward is not directly applied to the agent but to the object. The agent only gets reward when the object is moved.
> > > That's why PPO cannot solve this task effectively due to most of the movement of the agent is not rewarded and agent need to move a long distance to get the reward, especially when switching the contact point.
> > > However, we totally agree that the definition of sparse reward could be vague and we will clarify this as hard-to-explore reward in the revised version.

---

> ### Comment · Reviewer_X3LL · 2024-08-13
> **Thank you for the clarifications**
>
> Thank you for the clarifications.
>
> I have a question regarding the notation. Action $u_t$ at state $x_t$ leads to state $x_{t+1}$. Thus, the lhs of equation 1a should have $J_{x_{1:T}, u_{0:T-1}}$ instead of $J_{x_{1:T}, u_{1:T}}$? And I think $Y$ should also be modified to include $u_0$?

---

> ### Author Response · Authors · 2024-08-13
> **Thanks for the feedback**
>
> Yes, you are right.
> The optimization variable should be $u_{0:T-1}, x_{1:T}$ instead of $u_{1:T}, x_{1:T}$.
> Thank you so much for point it out! We will correct the notation in the revised version.

---

### Official Review · Reviewer_sfi9 · 2024-07-13

**Soundness:** 3
**Presentation:** 3
**Contribution:** 3
**Rating:** 7
**Confidence:** 3

**Summary:**

The paper presents Model-based Diffusion (MBD), a framework to train diffusion models to solve trajectory optimization (TO)  problems. In particular, the method assumes that the cost function associated with a TO problem can be evaluated for any trajectory of decision variables (states and control variables), which it then leverages to compute the target distribution of optimal solutions ($p_0(Y^{(0)})$) up to a normalizing constant. The denoising process in MBD performs a gradient ascent on intermediate distributions which are smoothed, easier to optimize versions of $p_0(Y^{(0)})$.

The method deals with unconstrained and constrained TO problems and is also able to incorporate demonstrations. MBD outperforms sota RL and also  sampling-based TO methods  in contact-rich tasks.

**Strengths:**

- The proposed method is evaluated on contact-rich, high-dimensional tasks and outperforms the baselines in terms of reward while keeping a computational time similar to the fastest baselines.

- The paper is well-written and the main ideas are clear and well-motivated. Furthermore, the method shows interesting connections to multi-stage optimization and also to the Cross Entropy Method for sample-based optimization.

- The method does not have many hyperparameters and they are robust across most tasks.

**Weaknesses:**

- While the comparison against RL is valuable and gives insights into the overall performance of MDB, the paper should better emphasize that the rewards reported in Table 2 are obtained by rolling out RL policies in an online closed-loop fashion, while the reward associated with MBD is computed according to an open loop execution.  Given that the paper lists the adaptation of MDB to online setup as future work, and that the rewards are computed in different ways, simply stating that MDB outperforms PPO by 59% can be misleading.  Qualifying such statements with an explanation of the different ways that rewards are computed would further improve the quality of the submission.

[Minor]
Typo L213 esitimation -> estimation

**Questions:**

- What are the challenges of adapting your method to online tasks with receding horizon strategies?  Is it a matter of training MBD to solve TO problems starting from more initial states?

-Once MDB has been trained, how long does it take to run the denoising process to get a new trajectory?

**Limitations:**

Limitations of the method are mentioned in the paper as potential lines of future work.

---

> ### Author Rebuttal · Authors · 2024-08-07
>
> We thank reviewer for the comments and suggestions. We address the comments and suggestions as follows:
>
> ---
>
> > the paper should better emphasize that the rewards reported in Table 2 are obtained by rolling out RL policies in an online closed-loop fashion.
>
> Thanks for the suggestion.
> We will further clearify this difference in updated maunscript.
> For the reason why we use RL as baseline, please refer to general response 1 and 2 for the detailed explanation of comparing MBD with RL and extend it to receding horizon version.
> Overall, MBD still outperforms RL in long horizon setting and can be easily extended to online setting with receding horizon strategy.
>
> ---
>
> > What are the challenges of adapting your method to online tasks with receding horizon strategies? Is it a matter of training MBD to solve TO problems starting from more initial states?
>
> Please refer to general response 2 for further details about the extension of MBD to online setting.
> In short, there is no fundamental limitation to extending MBD to closed-loop control with a receding horizon thanks to our fully model-based nature.
>
> ---
>
> > Once MDB has been trained, how long does it take to run the denoising process to get a new trajectory?
>
> Please refer to general response 3 for the training process clearification.
> In short, MBD does not require training components since we can calculate the score explicitly with model but can be further improved with extra data.

---

> > ### Comment · Reviewer_sfi9 · 2024-08-09
> > **Thanks for the clarifications!**
> >
> > I have read the author's response and appreciate the additional experiments and clarifications provided.
> >
> > Regarding the extension to the online setup, it would be useful to provide the frequency at which the close-loop control can be executed. Table 3 reports a compute time of at least 17 seconds for MC-MBD, which is inadequate for close-loop control, especially for highly dynamic systems like the humanoids.
> >
> > Using the naive receding horizon MBD algorithm introduced by the authors, the overall compute time to obtain a new solution is probably much lower due to the use of single-step MBD and to the forward shifting of the trajectory to initialize the next optimization step. Reporting the possible close-loop rates of MBD would also be valuable for the readers.
> >
> > Thanks.

---

> ### Author Response · Authors · 2024-08-09
> **Thanks for the suggestion!**
>
> > Using the naive receding horizon MBD algorithm introduced by the authors, the overall compute time to obtain a new solution is probably much lower due to the use of single-step MBD and to the forward shifting of the trajectory to initialize the next optimization step. Reporting the possible close-loop rates of MBD would also be valuable for the readers.
>
> Thanks for the suggestion. Showing the computation time of online MBD is indeed valuable for the audience who interested in real-time applications.
> We tested the online frequency on RTX 4070Ti GPU and the results are shown in Table 5.
> We will also include the computation time of MBD in the updated manuscript.
>
> Table 5: Online Running Frequency of receding horizon MBD
> | Env | Frequency (Hz) |
> | --- | --- |
> | Hopper | 4.56 |
> | HalfCheetah | 4.51 |
> | Ant | 10.28 | 7.46 |
> | Walker2d | 3.49 |
> | Humanoid Standup | 6.82 |
> | Humanoid Running | 4.03 |
>
> Please note that the frequency is calculated under the assumption of solving the whole 50 steps TO problem without reduced model at each iteration, which quite different from the MPC approach proposed in [1] which only solve the reduced model.
> Besides, the mujoco code environment we used is not optimized for GPU, so the actual frequency could be higher with optimized environment. In our work we just use Brax as a simple and easy-to-use option. As the major computation time of MBD is spent on the forward dynamics simulation, it can be further improved by using more efficient physics engine. Moreover, MBD's online frequency can be further improved by introducing learning components including learned value function, policy, or dynamics model like TD-MPC [2].
>
> In practice, by using GPU-optimized Unitree Go2 environment provided by [mujoco_menagerie](https://github.com/google-deepmind/mujoco_menagerie), we can achieve $52.3$ Hz control for Go2 robot with 12 DoF in walking task with 20 steps prediction horizon.
>
> Lastly, since MBD is designed to be a TO solver rather than a real-time controller, the computation time is not the main concern. But we totally agree that it is valuable to further explore the how to bring MBD to real-time which can solve the whole dynamic system without reduction model at each iteration and MBD has already shown its potential in this direction.
>
> [1] D. Kim, J. Di Carlo, B. Katz, G. Bledt, and S. Kim, “Highly Dynamic Quadruped Locomotion via Whole-Body Impulse Control and Model Predictive Control,” Sep. 14, 2019, arXiv: arXiv:1909.06586.
> [2] N. Hansen, X. Wang, and H. Su, “Temporal Difference Learning for Model Predictive Control,” Jul. 19, 2022, arXiv: arXiv:2203.04955. doi: 10.48550/arXiv.2203.04955.

---

### Author Rebuttal · Authors · 2024-08-07

# General Response

We thank all reviewers for their valuable comments.
We want to make the following clarifications and improvements to the manuscript:

## 1. Clarify the Problem Setting when Comparing with RL

We include **PPO/SAC as a performance reference not as a baseline** because there is no existing TO method that can solve such high-dimensional discontinuous tasks as we have shown in the experiments.
Model-free RL, especially PPO/SAC, is widely used in such tasks and is considered the SOTA method.

As pointed out by reviewer `sfi931`, RL differs from MBD in terms of **optimization objectives** (the horizon difference) and **optimization variables** (optimize for a policy v.s. a trajectory). We address the concern as follows:

**optimization objective**: To clarify the effect of different objectives, especially the horizon difference, we conducted an ablation study by swapping the optimization objectives of MBD and RL. RL optimizes for a longer horizon discounted reward $J = \sum_{t=0}^{H_\text{RL}} \gamma^t r_t, H=1000, \gamma<1$ while MBD optimizes for a shorter horizon undiscounted cumulative rewards $J = \sum_{t=0}^{H_\text{MBD}} r_t, H=50$, $\gamma=1$. Here we compare the performance of MBD and RL under each other's optimization objectives:

Table1: Mean Reward Across All Tasks of MBD and RL under TO and RL objectives
| Algo.| RL  |  MBD |
| - | - | - |
| RL Objective ($H=1000, \gamma<1$) |  1.33   | 1.93  |
| TO Objective ($H=50, \gamma=1$) |   0.23  | 2.12  |

As shown in Table 1 and Figure 1 in the accompanying PDF,
MBD outperforms RL by 44.5% under the RL objective and 805.5% under the TO objective.
The results demonstrate that MBD's superior performance is attributed to its better diffusion-style iterative optimization process compared with RL's random exploration.

**optimization variables**: As mentioned by reviewer `sfi913` and `X3LL`, RL optimizes for an online policy while MBD optimizes for an offline trajectory. Actually, we can easily extend offline MBD to online settings with a receding horizon strategy. We will address that in the next section.

## 2. Extension to Closed-loop Control with Receding Horizon

MBD is designed to be a TO solver, whose output is a trajectory rather than a policy.
That's why we only evaluate the open-loop MBD which represents the trajectory optimization capability of MBD.

We fully acknowledge reviewer `sfi913` and `X3LL`'s concerns in extending MBD to online control.
In short, **there is no fundamental limitation to extending MBD to closed-loop control with a receding horizon thanks to our fully model-based nature**. That means MBD can be easily conditioned on each step's observation and recalculate the trajectory in a receding horizon manner. Here is the naive receding horizon MBD algorithm:

```
Alg: MBD with Receding Horizon
Init: Optimize trajectory $x_{0:T}, u_{0:T-1}$ with MBD
For t=0 to T-1:
    Observe the state x_t
    Optimize trajectory $x_{t:T+t+1}, u_{t:T+t}$ with single-step MBD
    Apply the first control input $u_t$ to the system
    Shift the trajectory forward to provide an initialization for next step $x_{t+1:T+t+2}, u_{t+1:T+t+1}$
End For
```

Table 2: Mean Reward Across All Tasks of MBD (open-loop), MBD (receding) and RL
| RL | MBD(receding) | MBD |
| - | - | - |
| 1.33 | 2.32 | 2.12 |

Thanks to the closed-loop control and online optimization,
the results in Table 2 and Figure 2 in the accompanying PDF show that **the most naive receding horizon MBD can further improve MBD's performance by 9.6%** compared to the open-loop MBD.

## 3. The Training Process Clarification of MBD

**TL;DR: MBD does not involve explicit training components but can be further improved with extra data**.

Although MBD and MFD share the same iterative refinement process, MBD's setting is fundamentally different from generative model while more similar to MPC, where the dynamic model and constraints are well-defined but solving the problem is challenging. This difference also leads to the online sampling-based score computation from the model without off-line approximating it with a neural network.

This clarification should address concerns raised by reviewers `YGTC` and `sfi913` regarding the model’s capability for conditional planning. As our method is entirely model-based, it can adapt to any feasible observation and optimize trajectories without additional data collection or model retraining.

Acknowledging reviewer `X3LL`'s points, we agree on the value of introducing a learned module into the framework.  A potential future direction could involve integrating learning dynamics and cost functions, akin to model-based RL. We intend to explore this as a future direction and will note it in the revised manuscript.


## 4. Robustness and Generalization of MBD

Even though MBD only generates open-loop trajectories, **MBD is also robust to modeling error thanks to its iterative refinement process following the backward process of diffusion**.
In answering reviewer `X3LL`'s question about the robustness of MBD, we compare the performance of MBD with RL algorithms under 0.05 stochastic control noise in Figure 3 in the accompanying PDF.

Table 3: Mean Reward Across All Tasks of MBD and RL under Noisy Setting
| Algo. | RL| MBD(receding) | MBD |
| - | - | - | - |
| w/o noise | 1.33 | 2.32 | 2.12 |
| w/ noise | 1.00 | 2.23 | 1.65 |

As shown in Table 3 and Figure 2 in the accompanying PDF, adding noise does not lead to catastrophic failure in MBD, while the receding horizon version of MBD still outperforms RL by $65.3\%$ across all tasks under the noisy setting.
By refining the trajectory iteratively, MBD can effectively handle the noise and modeling error, making it more robust than RL.
Actually, in practice, standard trajectory output can be further fine-tuned with the updated model with data, which could further improve the performance of MBD given imperfect models.

---

### Decision · Program_Chairs · 2024-09-25

**Decision:**

Accept (poster)

**Comment:**

### Summary
The paper introduces a method called Model-Based Diffusion (MBD) for solving trajectory optimization (TO) problems. Unlike traditional model-free diffusion approaches, MBD leverages explicit model information to compute the score function in the diffusion process. This allows it to efficiently solve TO problems without relying on pre-existing data. The method also has the flexibility to incorporate data of varying quality, enhancing its applicability in diverse tasks. Experimental results demonstrate that MBD surpasses state-of-the-art methods, including reinforcement learning and sampling-based optimization, particularly in complex tasks involving contact-rich environments. The paper also highlights MBD's ability to integrate incomplete or imperfect data, offering practical advantages over standard diffusion models.

### Decision
The paper is well-written, and the application of model-based diffusion for the trajectory optimization problems is novel and interesting. The reviewers initially had concerns about the paper on the applicability of the proposed method to real-world problems, experiments (in particular, comparison to simple RL baseline like PPO), and clarity of the theory section. However, I think during the rebuttal period, the authors have done a good job addressing most of these concerns. The paper was borderline. However, the approach is novel, and there are no critical problems, the concerns raised by the reviewers could be addressed with a minor modification of the paper. I recommend the authors to carefully read the points made by the reviewers and clarify those points for the camera-ready version of the paper.